psychology

COVID-19, morality, moral psychology, utilitarianism

**Author for correspondence:**
Joaquin Navajas
e-mail: joaquin.navajas@utdt.edu

# Moral responses to the COVID-19 crisis

Joaquin Navajas[1,2], Facundo Álvarez Heduan[3], Gerry Garbulsky[4], Enzo Tagliazucchi[2,5], Dan Ariely[6] and Mariano Sigman[1,2,7]

[1]Laboratorio de Neurociencia, Universidad Torcuato Di Tella, Av. Figueroa Alcorta 7350, Buenos Aires C1428BCW, Argentina
[2]National Scientific and Technical Research Council (CONICET), Godoy Cruz 2290, Buenos Aires C1425FQB, Argentina
[3]El Gato y la Caja, Teodoro García 2474, Buenos Aires C1426DMR, Argentina
[4]TED, Araoz 727, Buenos Aires C1414DPO, Argentina
[5]Departamento de Física, Universidad de Buenos Aires, Av. Intendente Guiraldes 2160, Buenos Aires C1428EGA, Argentina
[6]The Fuqua School of Business, Duke University, 100 Fuqua Drive, Durham, NC 27708, USA
[7]Facultad de Lenguas y Educación, Universidad Nebrija, Calle de Sta. Cruz de Marcenado 27, Madrid 28015, Spain

JN, 0000-0001-8765-037X

The COVID-19 pandemic has raised complex moral dilemmas that have been the subject of extensive public debate. Here, we study how people judge a set of controversial actions related to the crisis: relaxing data privacy standards to allow public control of the pandemic, forbidding public gatherings, denouncing a friend who violated COVID-19 protocols, prioritizing younger over older patients when medical resources are scarce, and reducing animal rights to accelerate vaccine development. We collected acceptability judgements in an initial large-scale study with participants from 10 Latin American countries ($N = 15\,420$). A formal analysis of the intrinsic correlations between responses to different dilemmas revealed that judgements were organized in two dimensions: one that reflects a focus on human life expectancy and one that cares about the health of all sentient lives in an equitable manner. These stereotyped patterns of responses were stronger in people who endorsed utilitarian decisions in a standardized scale. A second pre-registered study performed in the USA ($N = 1300$) confirmed the replicability of these findings. Finally, we show how the prioritization of public health correlated with several contextual, personality and demographic factors. Overall, this research sheds light on the relationship between utilitarian decision-making and moral responses to the COVID-19 crisis.

# 1. Introduction

Over the last years, the psychological foundations of moral decision-making have been extensively studied by looking at how people react to abstract dilemmas [1–7]. For example, in trolley-type scenarios, participants are asked to decide whether it is acceptable to perform an action that will save the lives of some people at the cost of killing others [1]. While remarkably informative, these theoretical scenarios have several limitations. For example, people might be imprecise at estimating their future emotional states [8–11] and, as a consequence, they may be inaccurate at forecasting how they will judge and react when moral dilemmas are presented in the heat of the moment [12–15].

The COVID-19 pandemic has posed the world an extraordinarily difficult challenge that has brought moral dilemmas to the public sphere. Issues such as how to assign scarce medical resources (e.g. ventilators in cases of limited availability [16,17]), or whether it is acceptable to share sensitive private data to effectively trace the virus [18], or the tension between ensuring physical distancing or allowing economic and social activities [19], have shifted moral reasoning from theoretical to practical considerations and became part of public deliberations all the way from lay people to policy makers.

This results in an unprecedented opportunity to understand how people make moral decisions, which are relevant to policy making, in the midst of a healthcare crisis. Previous research has shown that moral dilemmas can help provide a graded notion of values across peoples and cultures [20]. But, above and beyond quantifying average preferences, here we aimed at performing a formal analysis of the intrinsic correlations between different moral dilemmas. Summarizing, the main objective of this work is to understand the organization of moral preferences in the emergence of real-life discussions about the COVID-19 crisis and to unfold their relationship with moral responses in classic dilemmas.

We ground our work on the study of utilitarian decision-making [21]. Utilitarianism is a normative theory that advocates that decisions ought to maximize a utility function reflecting the happiness and well-being of all affected individuals. As an example, in trolley-like dilemmas where different decisions lead to outcomes that vary in the number of deaths, the utilitarian prescription is to select the option in which fewer people die. More recently, it has been argued that people making utilitarian decisions in trolley-type dilemmas may in fact be relying on different considerations that produce the same outcome [22]. According to this view, utilitarian preferences result from two very distinct aspects: a 'negative' dimension that reflects a permissive attitude towards *instrumental harm*, and a 'positive' dimension, called *impartial beneficence*, which reflects an unbiased concern for the wellbeing of all sentient lives. Previous research has shown that, while *instrumental harm* correlates with psychopathic tendencies, *impartial beneficence* is associated with higher empathic concern [22].

Here, we empirically tested the idea that utilitarian decisions may be organized in two different dimensions and asked whether and how they relate to moral judgements about the pandemic. To address this aim, we designed five scenarios that probe different ways of responding to contemporary dilemmas (figure 1a). The first three dilemmas pose a tension between public health and other values of wellbeing: (1) effectively tracing the virus to control its spread versus protecting sensitive personal data, (2) ensuring physical distancing by forbidding public gatherings and business operations versus allowing those activities before a vaccine is developed, and (3) notifying a COVID-19 protocol breach versus protecting a close friend from facing prison. The remaining two dilemmas ask whether one should give priority to some lives above others in public health. The fourth dilemma is about the assignment of ventilators in case of limited availability and asks whether all patients should be treated equally or if younger individuals should be prioritized. This scenario could be thought of as a real-life variant of a trolley dilemma where one has to choose between saving younger or older people [20]. The fifth dilemma poses a tension between respecting the rights of animals in medical research versus accelerating the development of a vaccine.

One way to characterize moral responses to these scenarios is by looking at majority or average opinions. However, one may go one step further and study the pattern of correlations between different judgements. For example, if a person thinks that younger patients should be prioritized in their access to ventilators, will this person be more likely to support restrictions on social and economic activity? To answer this question, and to identify how moral responses are organized across these scenarios, here we examine the dimensionality of acceptability judgements (figure 1b,c).

This analysis requires a substantial amount of data, much more than is typically obtained in classic psychological and cognitive experiments. To solve this, we capitalized on a program which we have developed over the last years to perform cognitive experiments with large audiences [23–27]. Through this program we generated a sample of more than 15 000 participants proceeding from

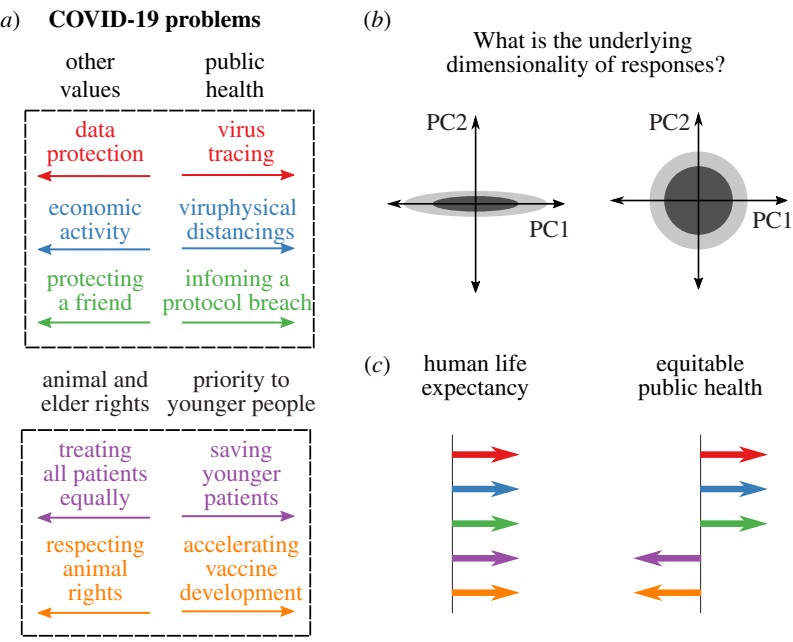

**Figure 1.** Moral problems about the COVID-19 pandemic and theoretical predictions. (*a*) We designed five moral scenarios that relate to problems of the COVID-19 outbreak (see Methods for full descriptions of each scenario). The first three pose a tension between prioritizing public health or other values and aspects of wellbeing. The last two ask if it is acceptable to prioritize younger human lives at the cost of breaking elder and animal rights. (*b*) What is the underlying structure of moral responses to these scenarios? In principle, if responses to all scenarios were highly correlated with each other, then they could be organized in one dimension. However, according to bi-dimensional theories of utilitarian decision-making, moral responses should be better explained by two components, each one correlating with a different dimension of utilitarianism (i.e. instrumental harm and impartial beneficence). (*c*) Dilemmas were conceived to examine two distinct focuses on public health. The first is a concern about maximizing human life expectancy whereby individuals prioritizing public health over other aspects of wellbeing also prioritize younger human lives over older and animal lives (left panel). The second hypothesis is a focus on equitable public health where prioritization of public health is compatible with respecting elder and animal rights (right panel).

10 Spanish-speaking Latin American countries (Study 1, $N = 15420$, 60.1% female, mean age: 32.0 y.o., range 18–95 years). We also tested the robustness of our findings in a second pre-registered study with a representative sample of the USA (Study 2, $N = 1300$, 48.3% female, mean age: 41.2 y.o., range 18–89 years).

Participants read the five scenarios about the COVID-19 pandemic in randomized order as well as the classic trolley problem in either its personal or impersonal version (see Methods for details). Each scenario described a clear action and participants reported three quantities using sliders: their perceived acceptability of that action (in a scale that ranged from 'completely wrong' to 'completely right'), their confidence in that judgement (from 'completely unsure' to 'completely sure') and the amount of distress that they think they would feel if they had to make a decision in that context (from 'none' to 'a great deal').

To study the pattern of correlations between different dilemmas, we examine the underlying dimensionality of the distribution of acceptability judgements. Responses to these dilemmas, which varied widely across the population (electronic supplementary material, figure S1), define a native five-dimensional space but, intuitively, if all points are scattered across a line, the underlying dimensionality of the data is one, if they are scattered within a plane, the dimensionality is two, and so on (figure 1*b*). While there are various techniques to formalize this intuition, one simple approach to statistically test the underlying dimensionality of the data is to perform a principal component analysis (PCA).

The advantage of this analysis, beyond its simplicity, is that it also provides the principal directions across which people tend to be aligned, which may allow us to identify—to a first degree of approximation—the argument of utility functions. Here, we consider and present two hypotheses about the relationship between utilitarian decision-making and moral responses to the pandemic. The first hypothesis is that people who make utilitarian judgements are focused on maximizing *human life expectancy*. If that were the case, then people prioritizing public health in the first three dilemmas should

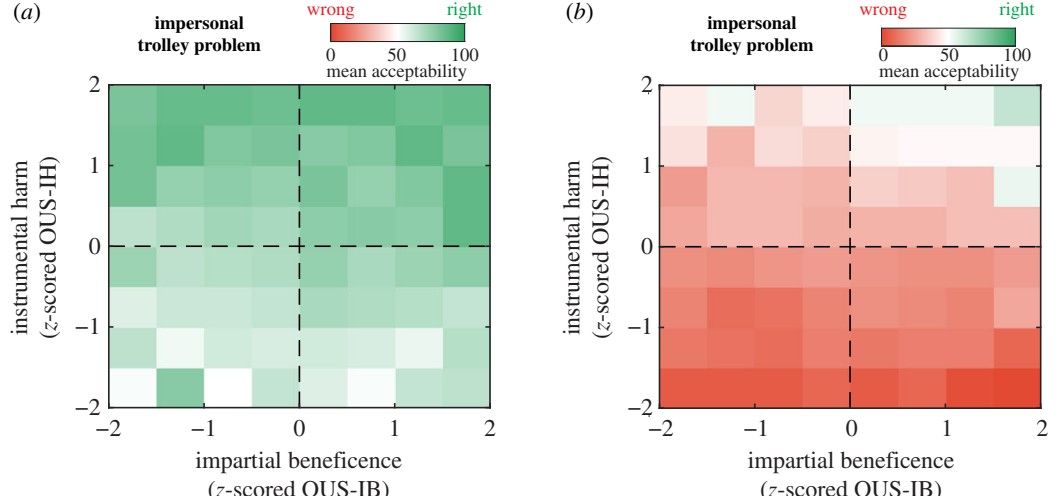

**Figure 2.** Instrumental harm and impartial beneficence as different dimensions of utilitarian decision-making. Images display the tendency to accept the utilitarian decision in the trolley problem, as a function of instrumental harm and impartial beneficence. Colours code the mean acceptability rating averaged across participants at a given level of z-scored OUS-IB (x-axis) and OUS-IH (y-axis). (a) Impersonal Trolley Problem. (b) Personal Trolley Problem.

also prioritize human over non-human animal life and younger over older patients (left panel of figure 1c). The second hypothesis is that utilitarian decision-making should correlate with a focus on *equitable public health*, i.e. people endorsing utilitarian principles in classic scenarios should prioritize public health in a fair and impartial manner contemplating and respecting all lives. The prototypical response predicted by this model is prioritizing public health in the first three dilemmas while respecting the rights of older patients and non-human animals in the last two scenarios (left panel of figure 1c).

## 2. Results

### 2.1. Individual differences in utilitarian decision-making

We measured individual differences in utilitarian decision-making [22] through a standardized scale (i.e. the Oxford Utilitarianism Scale, OUS, see Methods for details). This scale allows measuring individual variations in overall utilitarianism as well as in each of two dimensions: instrumental harm (OUS-IH) and impartial beneficence (OUS-IB). The study presenting this scale found a low degree of collinearity between both dimensions of utilitarianism (i.e. $r = 0.14$, $p < 0.01$, see Table 6 in [22]). Here, we also observed a low correlation between these two variables ($r = 0.13$, $p < 10^{-64}$) justifying the use of both dimensions as separate predictor variables ($R^2 = 0.01$, VIF = 1.01).

We also replicated a key finding of the original study: individuals with high scores on any of the two dimensions were more likely to support utilitarian decisions in the trolley dilemma (i.e. agree to sacrifice the life of a person to save a greater number of individuals) both in the impersonal (figure 2a, correlation with Instrumental Harm: $r = 0.26$, $p < 10^{-128}$, correlation with impartial beneficence: $r = 0.11$, $p < 10^{-21}$) as well as in the personal (figure 2b, correlation with Instrumental Harm: $r = 0.39$, $p < 10^{-281}$, correlation with impartial beneficence: $r = 0.10$, $p < 10^{-19}$) variants. Also consistent with that same previous study, we found that the correlation of utilitarian judgements in trolley-type dilemmas with instrumental harm was higher than the correlation with impartial beneficence (test for equal correlations, $z = 10.39$, $p < 10^{-200}$). Altogether, these results confirm that classic trolley dilemmas intermingle these two factors that contribute to utilitarian decisions.

The bi-dimensional theory of utilitarian psychology presented in that same previous study [22] makes two important predictions on other variables of the trolley problem which we collected in this large-scale sample. The first one relates to confidence: if impartial beneficence and instrumental harm are separate dimensions, then people's confidence should vary with the extremity of both two sub-scales. We observed strong evidence consistent with this prediction (bi-variate regression of absolute z-scored OUS sub-scales on confidence in the trolley problem, effect of instrumental harm: $\beta = 0.22 \pm 0.02$, $p < 10^{-28}$, effect of impartial beneficence: $\beta = 0.13 \pm 0.02$, $p < 10^{-11}$, see electronic supplementary material, figure S2).

This demonstrates that people scoring highly or lowly on instrumental harm or impartial beneficence displayed higher confidence in their judgements.

The second prediction relates to the self-reported level of distress about making these decisions. According to previous research, sub-clinical psychopathy has been shown to correlate positively with instrumental harm and, instead, empathic concern correlates positively with impartial beneficence but negatively with instrumental harm [22]. This predicts that self-reported distress will be greater for those people with more impartial beneficence and lesser for those with more instrumental harm. Observed values of self-reported distress about the trolley problem provided strong evidence supporting the bi-dimensional nature of both sub-scales (bi-variate regression of $z$-scored sub-scales on distress about the trolley problem, effect of instrumental harm: $\beta = -0.15 \pm 0.01$, $p < 10^{-41}$, effect of impartial beneficence: $\beta = 0.14 \pm 0.01$, $p < 10^{-35}$, see electronic supplementary material, figure S2).

## 2.2. Dimensionality analysis of moral problems of the pandemic

The overarching goal of this study is to understand the structure of moral responses to real-life dilemmas that have emerged during the COVID-19 crisis. We organize and narrow this general and broad aim by asking three questions that examine different fundamental fingerprints of the data: (1) What is the underlying dimensionality of the space of moral responses?; (2) What are the relevant principal components across which the data is organized?; and (3) How do these dimensions relate to individual differences in utilitarian decision-making? To answer these questions, we analysed the acceptability ratings that participants provided in response to the five moral scenarios about the COVID-19 pandemic (figure 3a).

The first question can be naturally addressed by investigating the number of significant principal components (i.e. the number of dimensions) that account for variance in the data above and beyond what would be expected by pure chance (see Methods). This analysis unveiled two main components explaining a significant proportion of the variance (figure 3b, random permutation test with 10 000 simulations, $p < 10^{-5}$), which is a mathematical way of reflecting that the embedding of moral responses is two-dimensional.

To address the second question, we looked at the directions of these two principal components, i.e. which combinations of responses across dilemmas explain large fractions of the variance in the data (electronic supplementary material, figure S3, information displayed in a PCA biplot [28] along with the joint probability distribution of the two components). This analysis revealed that the first principal component corresponds to a concern about *human life expectancy* (upper left panel of figure 3b), namely, a pattern of stereotyped responses where people prioritizing public health in the first three dilemmas also prioritize younger over older patients and human over non-human life. The second component is consistent with a focus on *equitable public health* (upper right panel of figure 3b) whereby prioritizing public health in the first three dilemmas is associated with the tendency to respect ethical procedures of animal research and the rights of elderly patients. These two principal components were still identifiable if we narrowed our analyses to four instead of five dilemmas (electronic supplementary material, figure S4). This robustness check suggests that the results presented so far are not driven by the observations made in any single dilemma but reflect a strong interplay of correlations across scenarios.

The third question asks whether these patterns of correlations match with the participants' tolerance to instrumental harm and their concern for impartial beneficence, as measured by the Oxford Utilitarianism Scale. To address this, we performed a regression of both components with the two sub-scales as predictor variables (figure 3c,d). We found that the first principal component (figure 3c; electronic supplementary material, figure S5) was significantly modulated by instrumental harm ($\beta = 0.31 \pm 0.04$, $p < 10^{-12}$) but not by impartial beneficence ($\beta = 0.02 \pm 0.03$, $p = 0.64$). Conversely, the second principal component (figure 3d; electronic supplementary material, figure S5) was explained by individual variations in impartial beneficence ($\beta = 0.24 \pm 0.03$, $p < 10^{-12}$) but not in instrumental harm ($\beta = -0.05 \pm 0.04$, $p = 0.23$). These two findings are consistent with the prediction that people who make utilitarian judgements may react differently to dilemmas which set priorities to some or all lives. Individuals with high scores of impartial beneficence will generally prefer not to assign priorities on ventilator use and will not agree on lowering the threshold of animal rights to accelerating the development of a vaccine, while the opposite choices will be preferred by those scoring highly on instrumental harm. Our results also reveal that both utilitarian dimensions agree on prioritizing public health over other non-health related aspects of wellbeing such as data privacy, economic activity and friendship (i.e. the first three scenarios in figure 1a).

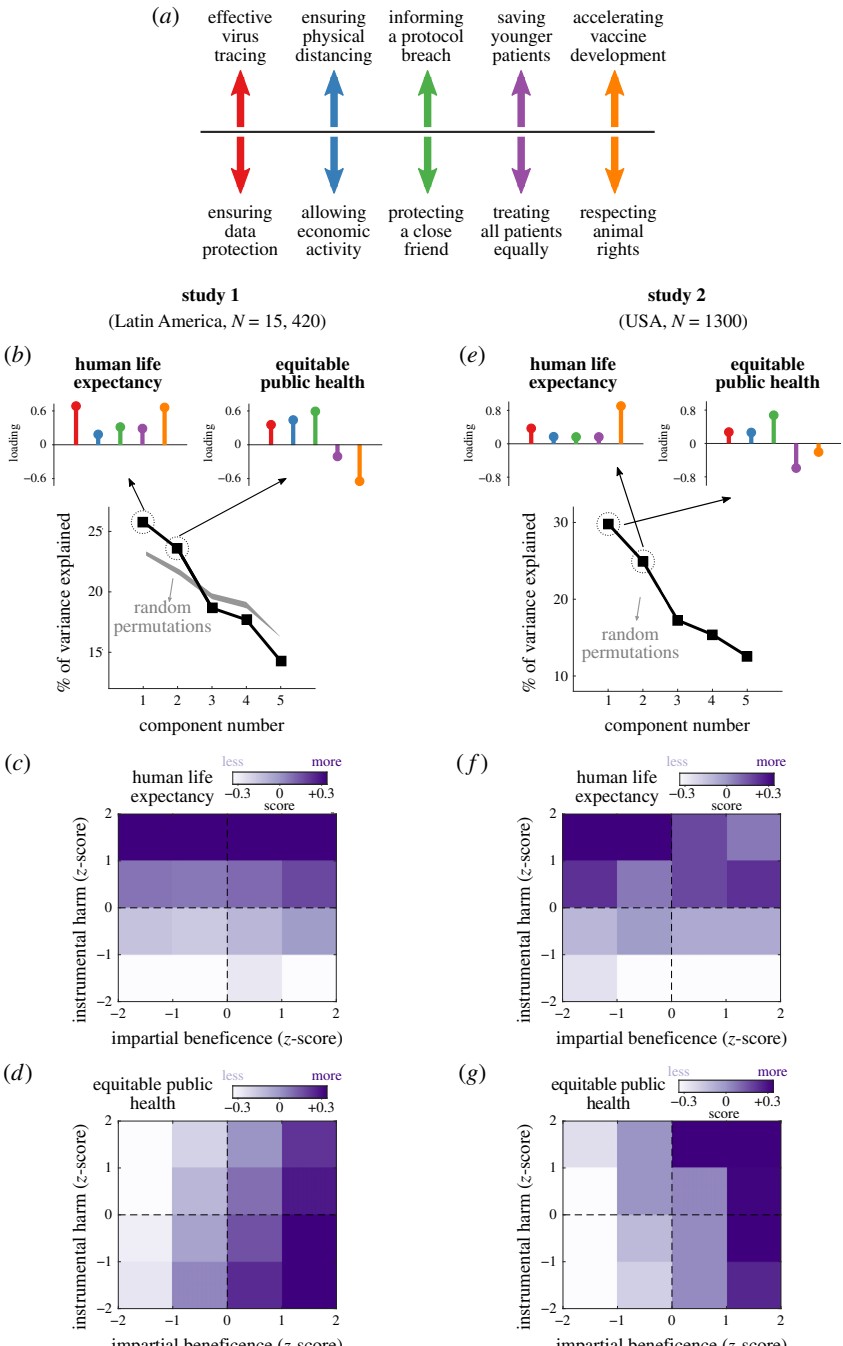

**Figure 3.** Dimensionality analysis of moral problems of the pandemic and relationship with utilitarian decision-making. (*a*) We performed a principal component analysis (PCA) of the acceptability ratings provided by participants in response to five moral dilemmas about the COVID-19 crisis. The arrows indicate how we coded the loadings of the principal components. For example, a positive loading in red reflects more agreement with public officials using private data to effectively trace the virus. (*b*) In Study 1 (performed in Latin America with 15420 participants), we observed that the first two principal components explained a significant amount of variance in the data. Black squares show the percentage of variance explained by each of the five principal components, and insets display the loadings of each variable, with each colour coding a different dilemma as in (*a*). The first principal component is consistent with a concern about human life expectancy and the second component with a focus on equitable public health. The light grey shade shows the ranges of variance percentages explained by 10 000 random permutations of the data. (*c*) Relationship between different dimensions of utilitarian decision-making and the first principal component (i.e. human life expectancy). Images show the average score of each component for participants with a given value of z-scored impartial beneficence (OUS-IB, x-axis) and instrumental harm (OUS-IH, y-axis). These data suggest that the principal components reflecting human life expectancy show greater correlation with instrumental harm than with impartial beneficence. (*d*) Same as (*c*) but for the second principal component. The data suggests that the principal components reflecting equitable public health show greater correlation with impartial beneficence than with instrumental harm. (*e,f*) Same as (*b–d*) but for the pre-registered replication study performed in the USA (Study 2).

## 2.3. Pre-registered replication study in the USA

One potential concern with Study 1 is that we could not control the representativeness of the sample that we obtained. For this reason, we performed a second pre-registered study in a different setting (Study 2), using a sample that is representative of the US population in terms of age, gender and ethnicity (electronic supplementary material, table S1). To determine sample size, we performed a Monte Carlo power analysis based on the data collected in Study 1 (electronic supplementary material, figure S6). Based on this analysis, we recruited 1300 participants, yielding an estimated power of 83.8%.

A second fundamental motivation for conducting this study is that it allowed us to test the robustness and replicability of our findings. With this aim, we pre-registered the hypotheses and methods of this study, specifying that its aim was to replicate the three main findings observed in Study 1 (https://aspredicted. org/59kk7.pdf). First, we predicted the existence of two principal components explaining a significant proportion of the variance in the data (figure 3b). Second, we hypothesized that these two dimensions would reflect a concern for human life expectancy (figure 1c, i.e. a principal component with positive loadings for all five scenarios) and an interest in equitable public health (figure 1c, i.e. a principal component with positive loadings for the first three scenarios and negative loadings for the last two). Third, we predicted a positive correlation between instrumental harm and human life expectancy and a positive correlation between impartial beneficence and equitable public health (figure 3c,d).

We first focused on the prediction that only two dimensions would explain a significant proportion of the variance in the data. To formally test this, we performed a random permutation analysis. As per our pre-registration, we generated 10 000 surrogated datasets by randomly shuffling the participants' labels for each scenario. Applying PCA on each surrogate, we estimated the distribution of explained variances for each component under the null hypothesis that there are no correlations in the data. As in Study 1, we observed that only the first two principal components explained more variance in the data than what is expected by chance (figure 3e, random permutation test, $p < 10^{-5}$).

Our second prediction was that these two dimensions would reflect human life expectancy and equitable public health. We tested this hypothesis by examining the signs of the five loadings for the first two principal components. As predicted, we found that one of those two dimensions had positive loadings for all dilemmas (upper left panel of figure 3e, a dimension consistent with a concern for human life expectancy) and the other component had positive loadings for the first three dilemmas and negative loadings for the last two (upper right panel of figure 3e, a dimension which reflects an interest in equitable public health).

The only difference between these findings and the ones observed in Study 1 is that the order of the two principal components was reversed (i.e. equitable public health explained more variance than human life expectancy). However, it is important to remark that we explicitly did not make any pre-registered claim about which of the two dimensions would be more representative in the data. This is because we reasoned that the different contexts associated with both studies may switch on how people give priority to these dimensions. Instead, the prediction of our study was that, while this may vary, variance in the data would still be structured in these two underlying dimensions. This effect, which was first observed in Study 1 (figure 3b), was successfully replicated in Study 2 (figure 3e).

Our third and last prediction was that each of these two components should correlate with a different aspect of utilitarian decision-making. As predicted, we observed that the component reflecting a concern for human life expectancy was positively correlated with having a permissive attitude towards instrumental harm (figure 3f, $r = 0.20$, $p < 10^{-13}$). We also found a significant correlation between the component suggesting an interest in equitable public health and impartial beneficence (figure 3g, $r = 0.39$, $p < 10^{-49}$). Importantly, these correlations were positive and significantly higher than the opposite and corresponding associations, i.e. if we correlated human life expectancy with impartial beneficence (test for equal correlations, $z = 5.5$, $p < 10^{-6}$) or equitable public health with instrumental harm ($z = 4.8$, $p < 10^{-6}$).

Overall, these findings indicate that Study 2 successfully replicated the results obtained in Study 1 despite using a different sampling strategy and collecting data in a different language, country and moment of the pandemic. These results provide strong evidence that moral judgements in response to these five dilemmas are organized in two dimensions, which are well predicted by people's scores in impartial beneficence and instrumental harm.

## 2.4. Contextual and personality factors

Our next question was whether, above and beyond these effects, any residual variance in the data could be explained by contextual [9,12] and personality [29,30] factors that may shape real-life moral decisions at

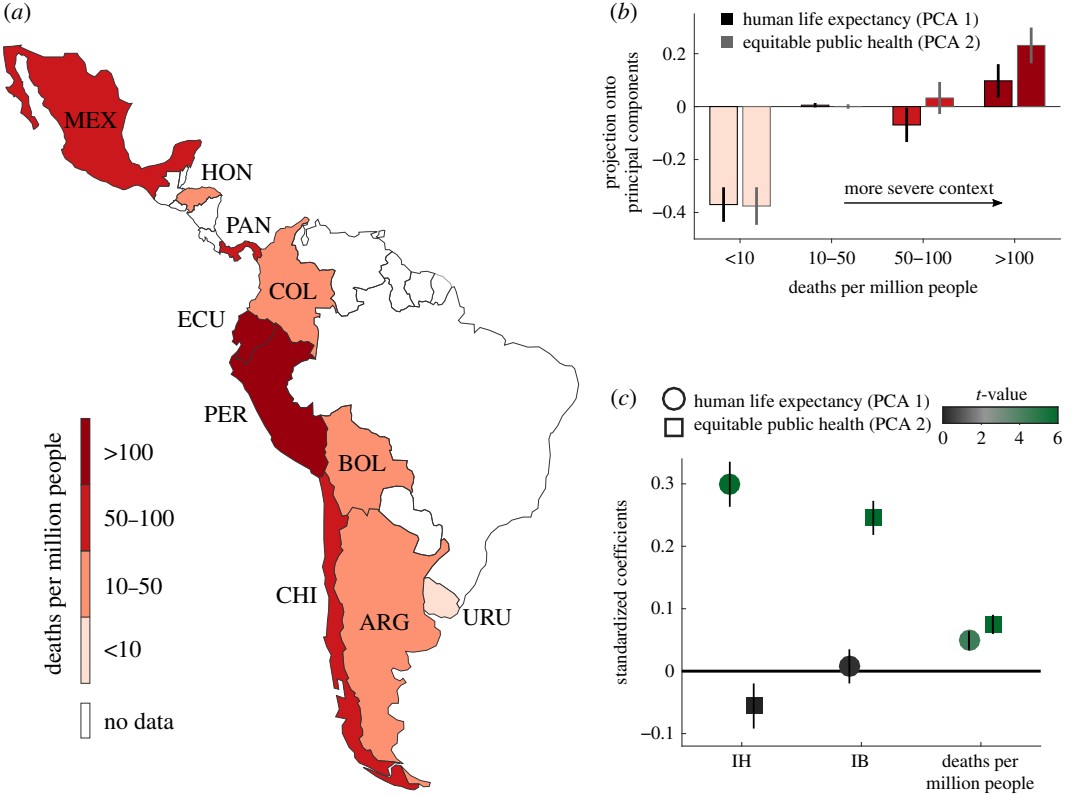

**Figure 4.** Country-level effects on moral preferences. (*a*) In Study 1, we collected data from 10 countries where the impact of the pandemic has been highly dissimilar at the time when the study was performed (May 2020). The map displays the countries from where we have obtained data. Colours show the *per capita* total number of confirmed COVID-19 deaths by the end of data collection. Country labels are Uruguay (URU), Argentina (ARG), Colombia (COL), Honduras (HOR), Bolivia (BOL), Chile (CHI), Mexico (MEX), Panama (PAN), Peru (PER) and Ecuador (ECU). (*b*) Projection onto the first two principal components (PCs) for participants proceeding from different countries. Participants were grouped based on the number of COVID-19 deaths per million people at the country from where they proceed. Less than 10: URU. Between 10 and 50: ARG, BOL, COL and HON. Between 50 and 100: CHI and MEX. More than 100: PER and ECU. Bars denote average PC score across participants, and vertical lines depict SEM. (*c*) We performed a multivariate regression analysis on the scores obtained for each PC. Circles show coefficient estimates for the first PC (Human Life Expectancy) and squares show coefficient estimates for the second PC (Equitable Public Health). Predictor variables displayed are Instrumental Harm (IH), Impartial Beneficence (IB) and the country-level pandemic's severity, quantified as the COVID-19 deaths per million people. Vertical lines show SEM and colours code the *t*-value associated with each coefficient. For the full list of predictor variables included in the analysis, see electronic supplementary material, tables S2 and S3.

constant values of utilitarian scores. This effort can be seen as a way of putting together different aspects of an individual—their tendency to make utilitarian decisions, their personality traits, and contextual elements—to disentangle how they differentially affect moral responses to the pandemic. To this end, we focused on the large-scale dataset obtained in Study 1 and ran a multivariate regression on the projection of each principal component as a function of 15 moral, contextual, personality and demographic variables (see Methods, electronic supplementary material, table S2 and table S3).

Previous research has suggested that negative contexts trigger affective states that could modify moral preferences [12], and so we reasoned that, even at constant values of utilitarianism, individuals proceeding from countries where the COVID-19 crisis is currently more severe could display greater preference to prioritize public health in moral dilemmas. To test this hypothesis, we analysed the data collected in Study 1 with participants proceeding from 10 Latin American countries (figure 4*a*). Importantly, by the time the study was performed (May 2020), the impact of the pandemic had shown a thirty-fold difference in confirmed COVID-19 deaths per million people (i.e. 6 for Uruguay and over 180 for Ecuador).

We observed that the intensity of the crisis, quantified by the *per capita* number of COVID-19 deaths, significantly modulated the participants' projection onto the first ($\beta = 0.05 \pm 0.01$, $p < 10^{-4}$) and second ($\beta = 0.07 \pm 0.01$, $p < 10^{-8}$) principal components (figure 4*b,c*). This association remained significant if, instead of using the *per capita* number of deaths, we used the *per capita* number of confirmed COVID-19 cases ($\beta = 0.03 \pm 0.01$, $p = 0.002$ for the first principal component, $\beta = 0.04 \pm 0.01$, $p = 0.001$ for

the second principal component). Overall, this analysis suggests that variations in the severity of the context modulated the projection onto both principal components.

However, the correlation of COVID-19 deaths (or any other variable, of course) with a principal component does not imply directly a correlation with responses to individual dilemmas. Just to give a simple example: if 4 of the 5 dilemmas would correlate positively with COVID-19 deaths then a principal component with all positive loadings (like the first principal component in Study 1) may also show a significant correlation although one of the dilemmas that defined this component did not correlate with COVID-19 deaths. Therefore, to understand the relationship between the severity of the pandemic and people's responses to individual dilemmas, we performed a *post hoc* correlation analysis. We found that the correlation between the number of COVID-19 deaths and the acceptability ratings to the first three scenarios was significantly positive (Scenario 1: $r = 0.03$, $p = 0.001$; Scenario 2: $r = 0.03$, $p = 0.001$; Scenario 3: $r = 0.04$, $p = 7 \times 10^{-7}$) while we observed a non-significant correlation with the responses to the last two scenarios (Scenario 4: $r = 0.01$, $p = 0.14$; Scenario 5: $r = 0.003$, $p = 0.63$). This finding suggests that the observed modulation given by the severity of the pandemic on the two principal components is driven by the first three scenarios, which have positive loadings for both components. In other words, the association between the severity of the pandemic and the two principal components is a result of a stronger prioritization of public health over other aspects of wellbeing and is unrelated to the trade-off between younger and older patients (Scenario 4) or between human and animal rights (Scenario 5).

To further examine the potential role played by contextual factors, we asked participants to report their personal proximity to the pandemic by indicating whether they were diagnosed with COVID-19 and/or if they knew someone who did. In our sample, 0.6% ($n = 92$) reported to have tested positive for COVID-19, 31.5% ($n = 4949$ participants) reported to have at least one acquaintance who tested positive, and 67.9% ($n = 10\,685$ participants) reported not knowing anyone diagnosed with the virus. We found no evidence that being diagnosed with COVID-19 (first principal component: $\beta = -0.06 \pm 0.05$, $p = 0.24$, second principal component: $\beta = -0.01 \pm 0.05$, $p = 0.73$), having COVID-like symptoms (first principal component: $\beta = 0.01 \pm 0.01$, $p = 0.39$, second principal component: $\beta = 0.004 \pm 0.009$, $p = 0.66$), or being close to patients who tested positive (first principal component: $\beta = -0.54 \pm 0.40$, $p = 0.17$, second principal component: $\beta = 0.09 \pm 0.40$, $p = 0.81$) modulated the projection to any of the two principal components. This indicates that the observed country-level differences are probably driven by concerns about the societal impact of the pandemic rather than the personal proximity to the virus.

We also reasoned that differences in general personality traits could also be associated with variability on how individuals make moral judgements. We asked participants to complete the Big Five Inventory [31] (see Methods) and investigated whether variations across these five dimensions of personality could explain residual variance in the data after controlling for the already described contextual and moral effects. We found that both principal components correlated positively with conscientiousness (first PC: $\beta = 0.03 \pm 0.01$, $p = 0.002$; second PC: $\beta = 0.03 \pm 0.01$, $p = 0.01$) and neuroticism (first PC: $\beta = 0.04 \pm 0.01$, $p = 0.003$; second PC: $\beta = 0.05 \pm 0.01$, $p < 10^{-4}$). This suggests that the tendency to prioritize public health over other aspects of wellbeing (either by focusing on human life expectancy or equitable public health) is especially strong in individuals who are diligent and responsible, but also susceptible to experience negative emotions that are prevalent during the pandemic like anxiety [32] and fear [33]. We also found that agreeableness was not associated with any of the two components, a result which might be explained by the fact that this trait strongly correlates with instrumental harm (negative correlation, $r = -0.09$, $p < 10^{-10}$) and impartial beneficence (positive correlation, $r = 0.10$, $p < 10^{-13}$). We again emphasize that these results reflect personality modulations on moral predispositions above and beyond the effects given by individual differences utilitarian scores.

Two personality traits, extraversion and openness, showed different effects on the two principal components, which differ on whether it is acceptable to prioritize younger people over older and animal lives (figure 1c). More extraversion was associated with higher scores in the first principal component and lower scores in the second (first PC: $\beta = 0.05 \pm 0.01$, $p < 10^{-4}$; second PC: $\beta = -0.06 \pm 0.01$, $p < 10^{-6}$). This is in line with previous observations that high extraversion is associated with psychopathic tendencies [34] and the propensity to break rules [35]. Openness showed a strong negative correlation with the first, but not the second, principal component (first PC: $\beta = -0.08 \pm 0.01$, $p < 10^{-13}$; second PC: $\beta = -0.02 \pm 0.01$, $p = 0.05$) which is consistent with research showing that scoring lowly on this trait is associated with the endorsement of different forms of prejudice, including ageism [36] and speciesism [37].

# 3. Discussion

The analyses presented in this study suggest that inter-individual differences in utilitarian decision-making predict judgements about moral problems of the COVID-19 crisis. These effects are robust to the inclusion of a long list of control variables (see electronic supplementary material, tables S2 and S3 for details) that index individual differences in contextual, personality and demographic factors. Moreover, the bi-dimensional organization of moral responses to the pandemic and their correlation with instrumental harm and impartial beneficence is consistent with a recently proposed theory of utilitarian decision-making [22].

While this paper focuses on the relationship between responses to moral problems of the COVID-19 crisis and utilitarian judgements, this does not mean that other theorized processes (for example, deontological mechanisms based on emotion or motivation to avoid harm [38,39]) do not exist or influence judgements above and beyond the observed effects [40,41]. Future research should explore how the endorsement of deontological considerations relate to moral responses about this healthcare crisis [42,43].

This research aimed at studying inter-individual differences in moral decision-making during the pandemic. We show that people who support utilitarian decisions in the trolley dilemma [1] and in the scenarios constituting the OUS [22] concurrently make a series of acceptability judgements in moral scenarios about the COVID-19 crisis. However, lay people arrive at judgements through a multiplicity of processes which may or may not coincide with the utilitarian principles proposed by theorists. Therefore, seeing that instrumental harm or impartial beneficence correlates with the two principal components observed in this study does not necessarily imply that participants always rely on those arguments across all scenarios or that the OUS scale will explain moral decisions in a wider set of dilemmas about the COVID-19 crisis [42]. Nonetheless, the observed correlation between those scores and judgements in the five tested scenarios is a robust phenomenon that is not explained by responses to any single dilemma (electronic supplementary material, figure S4) nor by the sampling strategy used in the initial study (figure 3*f,g*).

In Study 1, we generated a large-scale dataset of moral preferences in Latin America, a region that was seriously impacted by the pandemic, and that had contributed to more than 30% of the COVID-19 confirmed deaths by the time the study was performed. Our findings suggest that the crisis may have shifted the focus of utilitarian judgement (generally concerned about several dimensions of wellbeing) to setting a clear priority on public health, given that participants from countries where the pandemic had a more severe impact had a higher projection onto the two principal components. While the evidence presented here is correlational, this study represents a first step towards understanding country-level differences in policy preferences while controlling for individual-level variations in moral and personality variables, among others.

In Study 2, we showed that the utilitarian prioritization of public health over other aspects of wellbeing is a robust empirical observation. This is despite the replication study using a different sampling strategy and being performed in a different country, language and moment of the pandemic. While these data show that our main results are replicable in a very different setting, further research is needed to assert whether the utilitarian focus on public health is currently present in cross-cultural samples including other regions of the World [44,45]. In this sense, we believe that the methodology described here can be scaled up to performing large-scale online behavioural studies based on big data obtained from social media [46,47].

In conclusion, this research organized people's judgements about contemporary problems of the pandemic according to well-established utilitarian principles such as people's tolerance to instrumental harm and their concern for impartial beneficence. This classification should be an important input for policymakers aiming at constructing policies that represent their citizens' preferences. On top of that, we believe that understanding people's moral judgements about these issues can help political leaders to design more persuasive messages and improve the communication of public policies to address the crisis [48,49].

# 4. Methods

## 4.1. Context

Study 1 was part of an initiative called *TEDxperiments* aimed at constructing knowledge on human cognition by performing behavioural experiments with large audiences (www.tedxriodelaplata.org/

tedxperiments). Previous editions studied the use of a competition bias in a 'zero-sum fallacy' game [23], the role of deliberation in the wisdom of crowds [26], and the factors underlying consensus in polarized moral debates [27]. The study was in collaboration with *El Gato y la Caja*, a science popularization project that has previously performed large-scale behavioural studies on different aspects of human psychology [24,25]. On this occasion, the invitation to participate in the study was distributed online through the social media accounts of *TEDxperiments* and *El Gato y la Caja*. Participants signed a written informed consent, and were provided with contact details of the lead researcher. They were explicitly informed that their participation was voluntary and that they could withdraw from the study at any time. The procedure was approved by the ethics committee of CEMIC (Centro de Educación Médica e Investigaciones Clínicas Norberto Quirno) – Protocol 435, v. 5.

## 4.2. Participants

The invitation to perform Study 1 was first published on 6th May 2020 and we collected data until 31st May 2020. During this time window, we recruited $N = 15\,420$ participants (60.1% female, mean age: 32.0 y.o., range (18–95) years), most of them proceeding from Argentina ($n = 14\,443$). The remaining 977 participants were from Mexico ($n = 214$), Uruguay ($n = 204$), Peru ($n = 119$), Ecuador ($n = 107$), Bolivia ($n = 106$), Colombia ($n = 99$), Chile ($n = 92$), Honduras ($n = 37$) and Panama ($n = 32$).

## 4.3. Procedure

Participants read five scenarios related to the COVID-19 crisis in randomized order, as well as the trolley problem in either its personal or impersonal version. Half of the participants were presented with the trolley problem before reading the five scenarios about the pandemic, and the other half after. Each scenario described a clear action and we asked participants to report the acceptability of that action (in a scale from 'completely wrong' to 'completely right') and their confidence about the provided answer (from 'I feel completely unsure' to 'I feel completely sure'). We also asked them to imagine that they were the person responsible for deciding what to do in that situation and report the amount of distress that they think they would feel (from 'none' to 'a great deal'). In all cases, participants input their answers by moving three sliders on the screen, providing us with a value that ranged from 0 to 100. After reading the five scenarios about the pandemic and the trolley problem, participants completed the Oxford Utilitarianism Scale. Then, they were asked to report their age, gender, and whether had been diagnosed with COVID-19 and/or knew someone who had. Finally, they were invited to complete a standard personality test (see below for details).

## 4.4. Scenarios

The first scenario asks whether it is acceptable to allow the government to collect sensitive private data to allow tracking the virus. It reads: 'Suppose we can go back in time before the pandemic had exploded world wide. A new technology allows governments to automatically collect personal information about our movements and the people we have physical contact with. Governments can also have access to biometric data such as our temperature and respiratory frequency. This allows identifying people at risk of having COVID-19 and limiting the propagation of the virus by selecting who is quarantined and who is not. The government of the country where you live evaluates a plan to use this technology but understands that it does not protect the personal data of its citizens and decides not to approve it. Not approving this plan is…'.

The second scenario is about forbidding public gatherings, including several business activities, until a vaccine is found. It reads: 'The near future of our societies and how we will get out of the periods of strict confinement is unclear, among other things because there are great unknowns in the mutation rates of the virus, its seasonal variance, the probability of being re-infected, and the degree of immunity that we will reach. Societies are likely to create a compromise between minimizing health risks and preserving our desired lifestyles. Knowing this, a government decides to forbid all kinds of public gatherings (including restaurants, bars, concerts, cinemas and theatres) until a vaccine is found. The government's decision is…'. As a clarification, we stress that this scenario assumes that the development of a vaccine is equivalent to the end of the COVID-19 crisis. However, to date, there is no evidence on whether vaccines will be sufficiently effective to overcome the life-destroying effects of social activities. The aim of this scenario is to ask participants whether in the present situation (no vaccine) it is correct to forbid all public gatherings or not. This has been part of a very active public debate with very different attitudes and responses in different societies.

The third scenario deals with the tension between notifying a COVID-19 protocol breach versus protecting a close friend from facing prison. It reads: 'One of your close friends has high fever and was recently tested for COVID-19. The health authorities of your country have dictated that he should remain at home, in complete isolation, until the results are back. During this period, your friend went once to the supermarket to buy food. If the authorities find out about this, your friend might face prison and have a criminal record. You still do not know whether he will test positive for COVID-19 but decided to call the police and tell the authorities about this situation. Telling the authorities about this situation is…'.

The fourth scenario is about the assignment of scarce medical resources and asks whether younger people should be prioritized over older patients. It reads: 'Ventilators have become in many countries a limited resource and a medical bottleneck. An ethics committee is deciding how to set priorities on who will be treated and who will not. One option is to treat people on a first-come first-serve basis. A second argument is made that significantly younger patients (under 30 years old) should be prioritized and that, even when chances of recovery are considered to be the same, the younger patient should be treated before older patients. The ethics committee opts to implement the second option. This decision is…'

The fifth scenario poses a tension between animal rights and accelerating the development of a vaccine for COVID-19. It reads: 'A laboratory has all the necessary resources to develop a vaccine against coronavirus in only weeks. However, to do so, the researchers need to violate ethical guidelines for the use of animals in scientific research. Particularly, the experiments will cause excessive stress and pain to hundreds of rodents. The laboratory requests an exceptional permission to perform these experiments, but the government decides to reject this request. Rejecting this permission is…'

For the purposes of data analysis, we reverse-coded the acceptability ratings to the first and fifth dilemma, so that positive values indicate a prioritization to public health and a prioritization to saving younger human lives (figure 1*a*).

## 4.5. Trolley problems

The impersonal version of the trolley problem was presented as follows: 'A runaway train is speeding down the tracks towards five people who will be killed if the train continues on its present course. You are standing next to the tracks, but you are too far away to warn them. Next to you, there is a control switch that can redirect the train onto a different track, where there is only one person. If you decide to flip the control switch, that will provoke the death of one person, but the train will not continue its course and you will have saved the lives of five people. Flipping the control switch is …'.

The personal variant of the same classic scenario read: 'A runaway train is speeding down the tracks towards five people who will be killed if the train continues on its present course. You are standing next to the tracks, but you are too far away to warn them. Next to you, there is someone else and, if this person falls to the tracks, the train will derail. If you decide to push this person onto the tracks, that will provoke the death of this person, but the train will not continue its course and you will have saved the lives of five people. Pushing the person onto the tracks is…'

## 4.6. Utilitarianism scale

Individual differences in utilitarian tendencies were measured using the Oxford Utilitarianism Scale (OUS), which was developed and validated in a previous study [22]. The scale consists of nine items presented in randomized order, four of which measure permissive attitudes towards instrumental harm (OUS-IH, instrumental harm) and the remaining five evaluate people's impartial concern for the greater good (OUS-IB, impartial beneficence). All nine items were read on the same page and responses were collected using a 7-point scale from 'strongly disagree' to 'strongly agree'.

The four items of the OUS-IH sub-scale are: 1) 'It is morally right to harm an innocent person if harming them is a necessary means to helping several other innocent people'; 2) 'If the only way to ensure the overall well-being and happiness of the people is through the use of political oppression for a short, limited period, then political oppression should be used'; 3) 'It is permissible to torture an innocent person if this would be necessary to provide information to prevent a bomb going off that would kill hundreds of people'; 4) 'Sometimes it is morally necessary for innocent people to die as collateral damage—if more people are saved overall'.

The five items of the OUS-IB sub-scale are: 1) 'If the only way to save another person's life during an emergency is to sacrifice one's own leg, then one is morally required to make this sacrifice'; 2) 'From a moral point of view, we should feel obliged to give one of our kidneys to a person with kidney failure since we don't need two kidneys to survive, but really only one to be healthy'; 3) 'From a moral perspective, people should care about the well-being of all human beings on the planet equally; they should not favour the well-being of people who are especially close to them either physically or emotionally'; 4) 'It is just as wrong to fail to help someone as it is to actively harm them yourself'; and 5) 'It is morally wrong to keep money that one doesn't really need if one can donate it to causes that provide effective help to those who will benefit a great deal'.

## 4.7. Principal component analysis

We performed a PCA on the acceptability ratings across the five scenarios related to the COVID-19 pandemic. To identify the number of principal components that explained a significantly large proportion of the variance, we performed a random permutation analysis. We generated 10 000 surrogated datasets by randomly shuffling the participants' labels for each scenario. Applying PCA on each surrogate, we estimated the distribution of explained variances for the first to fifth component under the null hypothesis that there are no correlations in the data (grey shade in figure 3a). We then estimated the $p$-value of this statistical test as the fraction of times that a given component in the simulated data accounted for more variance than the one observed explained by that same component in the real data. We found that the first and second principal components explained more variance than any of the first and second components in the surrogated dataset, whereas the remaining three components explained less variance than was expected by pure chance.

## 4.8. Pre-registered replication study

Study 2 aimed at providing a high-powered pre-registered replication of Study 1 while using a sample which is representative of the United States population. To determine sample size, we performed a Monte Carlo power analysis based on the data collected in the original study. For each simulated sample size, we randomly sub-sampled the dataset 10 000 times without replacement and repeated the analyses performed in Study 1. We then estimated power as the fraction of times that we were able to successfully replicate three observations: (i) the existence of two principal components explaining a significant proportion of the variance in the data; (ii) the fact that these two dimensions would reflect human life expectancy and equitable public health (figure 1c); and (iii) the association between each principal component with instrumental harm and impartial beneficence. Based on the results of this power analysis (electronic supplementary material, figure S3), we decided to collect data from 1300 participants.

On 23rd October 2020, we pre-registered the hypotheses, design and analysis plan of Study 2 (https://aspredicted.org/59kk7.pdf). Data collection started on 23 October 2020 and proceeded until 30 October 2020. In this time window, we recruited the target sample size (aged 18–89, 48.3% female). This sample was obtained through Prolific, an online platform to recruit human participants for scientific research (https://www.prolific.co/). To obtain this sample, participants were stratified across three demographics: age, sex and ethnicity. Recruited participants were divided into subgroups that have similar proportions to the national population, based on data obtained from the US Census Bureau (https://www.census.gov/). The demographic characteristics of our sample and their comparison with the US population are available in electronic supplementary material, table S1.

## 4.9. Country-level COVID-19 data

Data about COVID-19 confirmed deaths and cases were obtained through a free and open-access platform called 'Our World in Data' that publishes research and data on some of the world's largest problems including health, poverty and human rights, among many others (https://ourworldindata.org/). Data about coronavirus published on that platform rely on figures issued by official authorities such as governments and ministries of health. The map displayed in figure 4a was first obtained through the website of 'Our World in Data' in .svg format and adapted to reflect custom data ranges.

## 4.10. Other individual-level contextual factors

In principle, one may reasonably ask whether showing COVID-like symptoms is a better predictor of moral responses than being diagnosed with COVID-19. This question is relevant given that a large fraction of carriers of COVID-19 are known to be asymptomatic. However, in our sample, the proportion of diagnosed individuals who were entirely asymptomatic is expected be very low, given that the criterion to get tested for COVID-19 in most Latin American countries is showing COVID-like symptoms. In Study 1, we asked participants to indicate whether they had fever. If they were diagnosed with COVID-19, we asked if that symptom showed up at some point during the course of the disease. If they had not been diagnosed with COVID-19, we asked if they had fever in the previous week. We observed that 94.7% of the participants who were diagnosed with COVID-19 also indicated to have had fever during the course of the disease. We also observed that 1.7% of our sample indicated having fever but not being diagnosed with COVID-19. To study whether symptoms could predict moral responses we added this binary variable as a predictor in the regression estimated in electronic supplementary material, tables S2 and S3. However, we found that it did not modulate any of the two first principal components (first principal component: $\beta = 0.01 \pm 0.01$, $p = 0.39$; second principal component: $\beta = 0.004 \pm 0.009$, $p = 0.66$). This finding is consistent with our previous observation that individual-level contextual variables did not modulate responses to the moral scenarios (electronic supplementary material, tables S2 and S3).

## 4.11. Individual and contextual effects on self-reported distress about COVID-19 problems

We observed that individuals scoring high on instrumental harm reported less distress about the trolley dilemma and the opposite effect was found for people with high scores on impartial beneficence (electronic supplementary material, figure S2). One question that arises from these findings is whether this effect holds for COVID-19 problems. To answer this question, we looked at the correlation between each sub-score of the Oxford Utilitarianism Scale and the average rating of self-reported distress across the five moral dilemmas about the pandemic. In accordance with our observations based on the Trolley Problem (electronic supplementary material, figure S2), we found that distress negatively correlated with instrumental harm ($r = -0.07$, $p = 4 \times 10^{-16}$) and positively correlated with impartial beneficence ($r = -0.15$, $p = 3 \times 10^{-73}$). We also asked whether contextual variables modulated self-reported measures of distress and found that the only significant predictor of less distress when considering moral problems of the pandemic was being diagnosed with COVID-19 ($\beta = -0.47 \pm 0.23$, $p = 0.03$).

## 4.12. Big five personality traits

In Study 1, we assessed personality using the 44-item Big Five Inventory [31], measuring participants on five dimensions of personality (extraversion, agreeableness, conscientiousness, neuroticism, openness). Each item is presented in a 5-choice answer format that ranges from complete disagreement (1 = very false for me) to complete agreement (5 = very true for me). We used a version in *Rioplatense* Spanish, adapted and validated in previous research [50].

## 4.13. Multivariate regression

To measure how the projection onto the first two principal components depended on moral, contextual, personality and demographic variables, we performed a multivariate linear regression. This regression included a total of 15 predictor variables, as displayed in electronic supplementary material, tables S2 and S3. Within those 15 regressors, we included two binary variables related to the structure of the study (i.e. 'experimental' variables): the order of presentation of the trolley problem (0: before the five main scenarios, 1: after reading the five main scenarios) and its version (0: impersonal, 1: personal).

Data accessibility. All data and codes to reproduce our findings [51] are available at https://osf.io/mwe9r/
Competing interests. The authors declare no competing interests.
Funding. This work was supported by the James McDonnell Foundation twenty-first Century Science Initiative in Understanding Human Cognition—Scholar Award (grant no. 220020334) and by a Sponsored Research Agreement between Facebook and Fundación Universidad Torcuato Di Tella (grant no. INB2376941).
Acknowledgements. We thank Victoria Milano, Rocco Di Tella, Juan Ignacio Cuile, Maria Agustina Nahas, Laura Gonzalez, Juan Manuel Garrido, Pablo Gonzalez, Nuria Caceres, Marco Sartorio and Lucia Freira for assistance with data collection.

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
