## [Peer Review File · Royal Society Open Science]

Review History

RSOS-210096.R0 (Original submission)

Review form: Reviewer 1 (Paul Conway)

Is the manuscript scientifically sound in its present form?

Yes

Are the interpretations and conclusions justified by the results?

No

Is the language acceptable?

No

Do you have any ethical concerns with this paper?

No

Have you any concerns about statistical analyses in this paper?

No

Recommendation?

Accept with minor revision (please list in comments)

Comments to the Author(s)

This paper, *Moral Reasoning about the COVID-19 Crisis*, presents two studies examining how scores on the 2D model of utilitarianism or OUS scale predict patterns of responses to covid dilemmas pitting different targets against one another. Overall, there is a lot to like about this paper. The topic is important and timely. The dilemmas are novel and interesting. The authors used a large sample and then a pre-registered replication. The findings are rather clear and make sense considering theory. Overall, I am positively disposed towards this paper and I would like to see a version published. My primary concerns are not with the core work but rather with the conceptual framing the authors have used. My hope is that with modest refinements to the phrasing in the intro and discussion this version could be published.

My theoretical concerns are several. First, the authors use the term 'moral reasoning' and claim they are studying 'reasoning mechanisms.' They're not the only ones in the field to use this type of terminology but it is incorrect and should be updated. This terminology comes from the philosophical origins of dilemma studies where philosophers assume that they are studying 'reasoning.' However, this approach assumes the mechanism rather than tests it, and appears to deny the possibility of humans making any decisions that don't involve reasoning. Many theorists describe mechanisms other than reasoning which influence moral judgments (e.g., Haidt, 2001; Greene, 2007) and researchers should not assume that they know what the mechanism is. Instead, I recommend using mechanism-neutral language. What they are really studying here is judgments. Whether such judgments result from reasoning or some other process or a combination of processes is a complex topic that is beyond the scope of this paper. By changing the term to moral judgments or responses or decisions, the authors can neatly sidestep the thorny question of whether it's truly reasoning or some other mechanism involved in driving dilemma responses.

Next, the authors describe what they call a 'unidimensional' model of moral judgment and they reference a paper by Joshua Greene. This is a strange argument to make because Greene's model is famously described as a dual-process model. The authors appear to describe judgments on these dilemmas as reflecting either greater or lesser utilitarian concerns as if there's no mechanism motivating deontological judgments. However, the core of this theory always was that one mechanism (e.g., reasoning about outcomes) motivates utilitarian judgments and a different mechanism (e.g., emotional aversion to harm) motivates deontological judgments. The model specifies that these two mechanisms clash and compete with one another to produce the ultimate response--hence, increasing emotionality can theoretically reduce willingness to cause sacrificial harm without changing the utilitarian outcomes of harm; likewise, increasing utilitarian outcomes of harm can theoretically increase sacrificial willingness even for the same aversive harmful action. The current paper is silent regarding the role of emotion or motivation to avoid causing harm, and this absence is misleading for a field that was founded on a dual-process model which is widely described and discussed, with substantial evidence (albeit may criticisms as well). I am not saying the authors must add additional mechanisms to their studies, but they should clarify the way they describe this model.

Part of the confusion may stem from the authors use of the OUS/2D utilitarian paper. Notably, that paper only describes mechanisms pertaining to utilitarian decision making and is silent regarding mechanisms that reflect deontological decision making--i.e., this approach is only focused on one of the two processes described by the dual-process model, and is silent regarding motivations to reject causing harm (i.e., actively make deontological judgments rather than simply avoid utilitarian judgments). In other words, the original model in the field is a dual-process model, and the 2D model further suggests that one of those processes may be further subdivided into more than one type of utilitarian consideration, but those does not mean that the other theorized process does not exist or influence judgments in its own right. Describing

traditional models as unidimensional appears to miss a large and growing literature that uses processing models to assess responses. The most prominent examples of this literature include process dissociation (Conway & Gawronski, 2013) and the CNI model (Gawronski et al, 2017). Papers using these techniques clearly demonstrate that more than one dimension contributes to what the authors here describe as a uni-dimensional model. In other words, what the authors describe as a single dimension with utilitarianism at one extreme and deontology at the other extreme is better described as the combination of one response tendencies to reject causing harm or follow moral rules and another response tendency to maximize overall outcomes--both elements in the OUS pertain only to the second dimension reflecting concern for outcomes. No dimension of the OUS maps theoretically onto deontological concerns about rejecting harm or consistently following moral rules.

I suggest the authors clarify these theory points as the way the paper is currently written may be confusing to readers. This paper is silent on the possibility that agreement with or endorsement of deontological considerations independent of the absence of utilitarian considerations may also play a role in driving judgments, and this point should be clarified for people who are familiar with the dual-process model and may wonder about the absence of reference to the second process in this model. Certainly, the authors should modify language about a uni-dimensional model as that does not accurately describe the model they refer to.

Another point that could stand clarification is the way the authors describe utilitarian and deontological principles in connection to the dilemmas in the paper. They do a nice job of briefly describing the relevant philosophical positions, but it should be clarified that responses on dilemmas should be conceptualized as distinct from endorsement of philosophical principles. In dilemmas, participants are making decisions and theorists may describe as consistent with utilitarian or philosophical principles, but it is not correct to say that the people making the decisions are necessarily invoking or endorsing those principles in the making of the decision. Hence, making a decision to accept sacrificial harm does not guarantee that one should be described as a utilitarian, as there are many reasons one may arrive at that judgment. Nor does making a judgment on one dilemma guarantee that one will make a similar judgment across many dilemmas, although there are somewhat stable individual differences. Likewise, one cannot determine whether someone counts as a deontologist or not based on responses to the two-dimensional OUS scale or dilemma responses, as there can be many factors that contribute to self-report responses--for example, psychopathy also predicts higher responses on instrumental harm, so a person endorsing this dimension may not necessarily qualify as 'utilitarian' even though some utilitarian philosophers also endorse the same items.

Conway and colleagues 2018 argued that the terms utilitarian and deontology better describe the decisions themselves rather than the people making those decisions, and recommend researchers avoid describing people as 'utilitarians or deontologists' except in rare cases involving practicing philosophers who fully appreciate and understand what it means to endorse these philosophical notions--it is more accurate to say 'people who make utilitarian judgments' or 'people accepting sacrificial harm' or 'utilitarian decision-makers' which implies the decision may be utilitarian without affirming the person as necessarily so. From this perspective, Figure 2 is misleading because it uses the terms 'utilitarians' and 'deontologists.'

In the case of the OUS, it may be reasonable to describe people as endorsing utilitarian principles, but that still doesn't mean that people should be described as 'utilitarians' full stop. People may endorse principles for a variety of reasons and the reasons they have as laypeople may not necessarily map onto the reasons why utilitarian expert philosophers endorse the same principles. The tendency to describe people as utilitarian or deontological obscures the true complexity of the many psychological processes contributing to judgments in this area. People arrive at judgments through a multiplicity of processes (including two studies in the current

work, but of course the current work omits measuring many other processes demonstrated in other work, such as affective reactions to harm, concerns about moral rules, general inaction, self-presentation, etc.).

Moreover, I don't think it's correct to describe people as 'deontologists' if they score low on one r both dimensions of the OUS. Instead, one would need a different measure asking questions directly about deontological ethics and how much people endorse those (e.g., see Robinson et al., 2015 who examine self-report items separately assessing utilitarian and deontological considerations). Failing to endorse utilitarian principles is not the same thing as definitely endorsing deontological principles.

Although an important theory issue for clarity, in a way these are also fairly modest wording issues and I hope the authors can simply tweak their language slightly without changing much of the core of the paper. The actual core contribution of this paper which is to distinguish between two different aspects of utilitarian thinking, to either maximize overall life expectancy or equitability in health treatment. This is where the main focus should lie, and cleaning up the theory language on the way to making this point will help the reader focus on and appreciate this main contribution to the literature rather than getting hung up on confusion regarding the way past work is conceptualized. I think it is interesting that impartial beneficence and instrumental harm demonstrate different relationships with maximizing life expectancy and equitable public healthcare access, and recommend the authors try to focus the discussion on this point itself without trying to make too many broader points about the overall nature of human moral decision-making given that doing so would require engaging with a much vaster literature than the current paper does--i.e., these findings do not rule out the presence or importance of many other factors such as concern for rules or affective decision-making and so on, they simply demonstrate the presence of two somewhat distinct influences of the 2D scale. I buy these findings, they make sense and the analyzes the author's present are convincing.

Review form: Reviewer 2 (Stephan Lewandowsky)

Is the manuscript scientifically sound in its present form?

Yes

Are the interpretations and conclusions justified by the results?

Yes

Is the language acceptable?

Yes

Do you have any ethical concerns with this paper?

No

Have you any concerns about statistical analyses in this paper?

Yes

Recommendation?

Major revision is needed (please make suggestions in comments)

Comments to the Author(s)

Review of MS RSOS-210096

by Navajas et al.

Reviewer: Stephan Lewandowsky

Summary and Overall Recommendation

The paper investigates how people process moral dilemmas relating to the COVID-19 pandemic. The pandemic opened up a slew of moral dilemmas relating to privacy, decisions about rationing medical care, and trading off the economy against public health. The authors ground their work in the classical distinction between utilitarian and deontological principles of morality, and they seek to tease apart those two dimensions by using 5 different dilemmas that are expected to elicit different response patterns depending on which deontological or utilitarian maxim (maximizing life expectancy vs. equitable public health) people choose to satisfy.

The results show (a) that in the conventional trolley problem questionnaire responses that correspond to those two dimensions are independent and can be related to the observed acceptability of the personal and impersonal version of the problem. (b) For moral dilemmas involving the pandemic, two dimensions were sufficient to capture the variance in the large sample ($N > 15K$) across the 5 dilemmas. The first dimension relates to concern about human life expectancy, whereas the second dimension relates to equity in health, both across people and even across species. (c) These two dimensions were found to be related to the two different underlying dimensions of utilitarian thinking. These findings were replicated in a preregistered study using a representative American sample.

In addition, an examination of individual-differences variables and demographic variables (e.g., severity of the crisis) revealed a number of interesting findings. For example, the severity of the pandemic was associated with endorsement of the “maximizing life expectancy” options whereas there was no correlation with the “equitable public health” dilemmas.

Turning to evaluation, there is much to like about the paper. It is generally well written, concise, easy to follow, and highly informative. The method is innovative but also very solid, and the data are analyzed competently and in great depth.

On balance, this paper makes a very interesting contribution and I am positively inclined towards publication. However, I believe that there is one major question that needs to be resolved by revision before I can wholeheartedly recommend publication. (I should add that I rarely only pick up on a single issue, so the authors are to be commended for submitting a terrific paper that is already in very good shape).

Major point

My biggest concern involves the identifiability of the two principal components, given that the first 3 dilemmas load positively on both. How robust are the observed findings given that ultimately only two dilemmas carry much of the diagnostic load? How robust are the findings in light of this? I am prepared to accept the results as presented, but I would ask for some assurance that there are no surprises lurking in the shadows. I'll leave it up to the authors to decide how to respond to this but I suspect most readers would like some more assurance.

Decision letter (RSOS-210096.R0)

Dear Dr Navajas

The Editors assigned to your paper RSOS-210096 "Moral reasoning about the COVID-19 crisis" have now received comments from reviewers and would like you to revise the paper in accordance with the reviewer comments and any comments from the Editors. Please note this decision does not guarantee eventual acceptance.

Please submit your revised manuscript and required files (see below) no later than 21 days from today's (ie 04-Jun-2021) date. Note: the ScholarOne system will 'lock' if submission of the revision is attempted 21 or more days after the deadline. If you do not think you will be able to meet this deadline please contact the editorial office immediately.

on behalf of Dr Giorgia Silani (Associate Editor) and Essi Viding (Subject Editor)
openscience@royalsociety.org

Associate Editor Comments to Author (Dr Giorgia Silani):

Comments to the Author:

I have now received the reviews of your manuscript referenced above. The reviewers are generally very positive about your work and clearly in favor of publication. However, they have also listed a number of suggestions which mainly include conceptual/theoretical additions.

These suggestions are outlined in their reviews which have been included below. I invite you to revise the manuscript accordingly.

Reviewer comments to Author:

Reviewer: 1

Comments to the Author(s)

This paper, Moral Reasoning about the COVID-19 Crisis, presents two studies examining how scores on the 2D model of utilitarianism or OUS scale predict patterns of responses to covid dilemmas pitting different targets against one another. Overall, there is a lot to like about this

paper. The topic is important and timely. The dilemmas are novel and interesting. The authors used a large sample and then a pre-registered replication. The findings are rather clear and make sense considering theory. Overall, I am positively disposed towards this paper and I would like to see a version published. My primary concerns are not with the core work but rather with the conceptual framing the authors have used. My hope is that with modest refinements to the phrasing in the intro and discussion this version could be published.

My theoretical concerns are several. First, the authors use the term 'moral reasoning' and claim they are studying 'reasoning mechanisms.' They're not the only ones in the field to use this type of terminology but it is incorrect and should be updated. This terminology comes from the philosophical origins of dilemma studies where philosophers assume that they are studying 'reasoning.' However, this approach assumes the mechanism rather than tests it, and appears to deny the possibility of humans making any decisions that don't involve reasoning. Many theorists describe mechanisms other than reasoning which influence moral judgments (e.g., Haidt, 2001; Greene, 2007) and researchers should not assume that they know what the mechanism is. Instead, I recommend using mechanism-neutral language. What they are really studying here is judgments. Whether such judgments result from reasoning or some other process or a combination of processes is a complex topic that is beyond the scope of this paper. By changing the term to moral judgments or responses or decisions, the authors can neatly sidestep the thorny question of whether it's truly reasoning or some other mechanism involved in driving dilemma responses.

Next, the authors describe what they call a 'unidimensional' model of moral judgment and they reference a paper by Joshua Greene. This is a strange argument to make because Greene's model is famously described as a dual-process model. The authors appear to describe judgments on these dilemmas as reflecting either greater or lesser utilitarian concerns as if there's no mechanism motivating deontological judgments. However, the core of this theory always was that one mechanism (e.g., reasoning about outcomes) motivates utilitarian judgments and a different mechanism (e.g., emotional aversion to harm) motivates deontological judgments. The model specifies that these two mechanisms clash and compete with one another to produce the ultimate response--hence, increasing emotionality can theoretically reduce willingness to cause sacrificial harm without changing the utilitarian outcomes of harm; likewise, increasing utilitarian outcomes of harm can theoretically increase sacrificial willingness even for the same aversive harmful action. The current paper is silent regarding the role of emotion or motivation to avoid causing harm, and this absence is misleading for a field that was founded on a dual-process model which is widely described and discussed, with substantial evidence (albeit may criticisms as well). I am not saying the authors must add additional mechanisms to their studies, but they should clarify the way they describe this model.

Part of the confusion may stem from the authors use of the OUS/2D utilitarian paper. Notably, that paper only describes mechanisms pertaining to utilitarian decision making and is silent regarding mechanisms that reflect deontological decision making--i.e., this approach is only focused on one of the two processes described by the dual-process model, and is silent regarding motivations to reject causing harm (i.e., actively make deontological judgments rather than simply avoid utilitarian judgments). In other words, the original model in the field is a dual-process model, and the 2D model further suggests that one of those processes may be further subdivided into more than one type of utilitarian consideration, but those does not mean that the other theorized process does not exist or influence judgments in its own right. Describing traditional models as unidimensional appears to miss a large and growing literature that uses processing models to assess responses. The most prominent examples of this literature include process dissociation (Conway & Gawronski, 2013) and the CNI model (Gawronski et al, 2017). Papers using these techniques clearly demonstrate that more than one dimension contributes to what the authors here describe as a uni-dimensional model. In other words, what the authors

describe as a single dimension with utilitarianism at one extreme and deontology at the other extreme is better described as the combination of one response tendencies to reject causing harm or follow moral rules and another response tendency to maximize overall outcomes--both elements in the OUS pertain only to the second dimension reflecting concern for outcomes. No dimension of the OUS maps theoretically onto deontological concerns about rejecting harm or consistently following moral rules.

I suggest the authors clarify these theory points as the way the paper is currently written may be confusing to readers. This paper is silent on the possibility that agreement with or endorsement of deontological considerations independent of the absence of utilitarian considerations may also play a role in driving judgments, and this point should be clarified for people who are familiar with the dual-process model and may wonder about the absence of reference to the second process in this model. Certainly, the authors should modify language about a uni-dimensional model as that does not accurately describe the model they refer to.

Another point that could stand clarification is the way the authors describe utilitarian and deontological principles in connection to the dilemmas in the paper. They do a nice job of briefly describing the relevant philosophical positions, but it should be clarified that responses on dilemmas should be conceptualized as distinct from endorsement of philosophical principles. In dilemmas, participants are making decisions and theorists may describe as consistent with utilitarian or philosophical principles, but it is not correct to say that the people making the decisions are necessarily invoking or endorsing those principles in the making of the decision. Hence, making a decision to accept sacrificial harm does not guarantee that one should be described as a utilitarian, as there are many reasons one may arrive at that judgment. Nor does making a judgment on one dilemma guarantee that one will make a similar judgment across many dilemmas, although there are somewhat stable individual differences. Likewise, one cannot determine whether someone counts as a deontologist or not based on responses to the two-dimensional OUS scale or dilemma responses, as there can be many factors that contribute to self-report responses--for example, psychopathy also predicts higher responses on instrumental harm, so a person endorsing this dimension may not necessarily qualify as 'utilitarian' even though some utilitarian philosophers also endorse the same items.

Conway and colleagues 2018 argued that the terms utilitarian and deontology better describe the decisions themselves rather than the people making those decisions, and recommend researchers avoid describing people as 'utilitarians or deontologists' except in rare cases involving practicing philosophers who fully appreciate and understand what it means to endorse these philosophical notions--it is more accurate to say 'people who make utilitarian judgments' or 'people accepting sacrificial harm' or 'utilitarian decision-makers' which implies the decision may be utilitarian without affirming the person as necessarily so. From this perspective, Figure 2 is misleading because it uses the terms 'utilitarians' and 'deontologists.'

In the case of the OUS, it may be reasonable to describe people as endorsing utilitarian principles, but that still doesn't mean that people should be described as 'utilitarians' full stop. People may endorse principles for a variety of reasons and the reasons they have as laypeople may not necessarily map onto the reasons why utilitarian expert philosophers endorse the same principles. The tendency to describe people as utilitarian or deontological obscures the true complexity of the many psychological processes contributing to judgments in this area. People arrive at judgments through a multiplicity of processes (including two studies in the current work, but of course the current work omits measuring many other processes demonstrated in other work, such as affective reactions to harm, concerns about moral rules, general inaction, self-presentation, etc.).

Moreover, I don't think it's correct to describe people as 'deontologists' if they score low on one or both dimensions of the OUS. Instead, one would need a different measure asking questions directly about deontological ethics and how much people endorse those (e.g., see Robinson et al., 2015 who examine self-report items separately assessing utilitarian and deontological considerations). Failing to endorse utilitarian principles is not the same thing as definitely endorsing deontological principles.

Although an important theory issue for clarity, in a way these are also fairly modest wording issues and I hope the authors can simply tweak their language slightly without changing much of the core of the paper. The actual core contribution of this paper which is to distinguish between two different aspects of utilitarian thinking, to either maximize overall life expectancy or equitability in health treatment. This is where the main focus should lie, and cleaning up the theory language on the way to making this point will help the reader focus on and appreciate this main contribution to the literature rather than getting hung up on confusion regarding the way past work is conceptualized. I think it is interesting that impartial beneficence and instrumental harm demonstrate different relationships with maximizing life expectancy and equitable public healthcare access, and recommend the authors try to focus the discussion on this point itself without trying to make too many broader points about the overall nature of human moral decision-making given that doing so would require engaging with a much vaster literature than the current paper does--i.e., these findings do not rule out the presence or importance of many other factors such as concern for rules or affective decision-making and so on, they simply demonstrate the presence of two somewhat distinct influences of the 2D scale. I buy these findings, they make sense and the analyses the author's present are convincing.

Reviewer: 2

Comments to the Author(s)

Review of MS RSOS-210096

by Navajas et al.

Reviewer: Stephan Lewandowsky

Summary and Overall Recommendation

The paper investigates how people process moral dilemmas relating to the COVID-19 pandemic. The pandemic opened up a slew of moral dilemmas relating to privacy, decisions about rationing medical care, and trading off the economy against public health. The authors ground their work in the classical distinction between utilitarian and deontological principles of morality, and they seek to tease apart those two dimensions by using 5 different dilemmas that are expected to elicit different response patterns depending on which deontological or utilitarian maxim (maximizing life expectancy vs. equitable public health) people choose to satisfy.

The results show (a) that in the conventional trolley problem questionnaire responses that correspond to those two dimensions are independent and can be related to the observed acceptability of the personal and impersonal version of the problem. (b) For moral dilemmas involving the pandemic, two dimensions were sufficient to capture the variance in the large sample ($N > 15K$) across the 5 dilemmas. The first dimension relates to concern about human life expectancy, whereas the second dimension relates to equity in health, both across people and even across species. (c) These two dimensions were found to be related to the two different underlying dimensions of utilitarian thinking. These findings were replicated in a preregistered study using a representative American sample.

In addition, an examination of individual-differences variables and demographic variables (e.g., severity of the crisis) revealed a number of interesting findings. For example, the severity of the pandemic was associated with endorsement of the "maximizing life expectancy" options whereas there was no correlation with the "equitable public health" dilemmas.

Turning to evaluation, there is much to like about the paper. It is generally well written, concise, easy to follow, and highly informative. The method is innovative but also very solid, and the data are analyzed competently and in great depth.

On balance, this paper makes a very interesting contribution and I am positively inclined towards publication. However, I believe that there is one major question that needs to be resolved by revision before I can wholeheartedly recommend publication. (I should add that I rarely only pick up on a single issue, so the authors are to be commended for submitting a terrific paper that is already in very good shape).

Major point

My biggest concern involves the identifiability of the two principal components, given that the first 3 dilemmas load positively on both. How robust are the observed findings given that ultimately only two dilemmas carry much of the diagnostic load? How robust are the findings in light of this? I am prepared to accept the results as presented, but I would ask for some assurance that there are no surprises lurking in the shadows. I'll leave it up to the authors to decide how to respond to this but I suspect most readers would like some more assurance.

===PREPARING YOUR MANUSCRIPT===

If you have been asked to revise the written English in your submission as a condition of publication, you must do so, and you are expected to provide evidence that you have received language editing support. The journal would prefer that you use a professional language editing service and provide a certificate of editing, but a signed letter from a colleague who is a native speaker of English is acceptable. Note the journal has arranged a number of discounts for authors using professional language editing services

(<https://royalsociety.org/journals/authors/benefits/language-editing/>).

===PREPARING YOUR REVISION IN SCHOLARONE===

Author's Response to Decision Letter for (RSOS-210096.R0)

See Appendix A.

RSOS-210096.R1 (Revision)

Review form: Reviewer 1 (Paul Conway)

Is the manuscript scientifically sound in its present form?

Yes

Are the interpretations and conclusions justified by the results?

Yes

Is the language acceptable?

Yes

Do you have any ethical concerns with this paper?

No

Have you any concerns about statistical analyses in this paper?

No

Recommendation?

Accept with minor revision (please list in comments)

Comments to the Author(s)

This paper, Moral responses to the COVID-19 crisis, presents a revised version of a manuscript examining how the OUS predicts responses to COVID dilemmas. I already saw much to like about the previous version of the manuscript, and I continue to do so. My concerns were primarily with the conceptual framing of the argument. Here I commend the authors for their thoughtful and thorough response to these concerns, as well as the concerns of the other reviewers. I find this version of the manuscript much improved, with much clearer terminology about decisions versus process, labeling people as decision-makers, and distinguishing between increased utilitarian and reduced deontological decision-making. I think a version of the manuscript very close to this one should be published, pending slight clarification, and I do not think I need to personally see the revision.

The authors write that “we now explicitly state that our work is silent regarding mechanisms that reflect deontological judgements.” While I appreciate the spirit of this argument, I suggest it is not entirely correct. Insofar as the authors examine dilemmas that arguably pit deontological against utilitarian concerns, responses arguably are exactly as informative about deontological as they are of utilitarian responding. In other words, any argument the authors could make that X increases utilitarian responding could simply be rephrased as ‘X reduced deontological responding.’ No meaningful distinction can be made between these statements without use of modelling approaches such as process dissociation or the CNI model. It might be helpful to slightly reword the caveats the authors have helpfully added to clarify this point.

Note this argument primarily applies to the portion of the results pertaining to ‘trolley dilemma’ responses. Arguably the OUS is designed to capture ‘utilitarian thinking’ (although there can be other interpretations as I noted in my previous review). Insofar as this is true, it is reasonable to argue that the authors are examining how ‘utilitarian thinking’ influences COVID responding. So it’s a slightly nuanced point, but arguably the dilemma responses could be reworded as assessing deontological responding and flipping the sign of the correlations and remain equally correct interpretations of the data. Note as well that I say ‘deontological responding’ here as the authors are correct that raw dilemma responses are ambiguous whether they truly assess either utilitarian or deontological ‘thinking’.

Overall, I comment the authors on a fine piece here. Finally, I must apologize as my review likely delayed their response from the editor. Unfortunately, this review found me in the midst of an international move which was especially complicated by COVID restrictions and further complicated by unexpected government problems requiring many hours sorting out immigration status and then active COVID cases in the household... so my reviewing efficiency has fallen off a cliff.

Review form: Reviewer 2 (Stephan Lewandowsky)

Is the manuscript scientifically sound in its present form?

Yes

Are the interpretations and conclusions justified by the results?

Yes

Is the language acceptable?

Yes

Do you have any ethical concerns with this paper?

No

Have you any concerns about statistical analyses in this paper?

No

Recommendation?

Accept as is

Comments to the Author(s)

Review of MS RSOS-210096.R1

by Navajas et al.

Reviewer: Stephan Lewandowsky

Summary and Overall Recommendation

The paper investigates how people process moral dilemmas relating to the COVID-19 pandemic. The pandemic opened up a slew of moral dilemmas relating to privacy, decisions about rationing medical care, and trading off the economy against public health. The authors ground their work in the classical distinction between utilitarian and deontological principles of morality, and they seek to tease apart those two dimensions by using 5 different dilemmas that are expected to elicit

different response patterns depending on which deontological or utilitarian maxim (maximizing life expectancy vs. equitable public health) people choose to satisfy.

The results show (a) that in the conventional trolley problem questionnaire responses that correspond to those two dimensions are independent and can be related to the observed acceptability of the personal and impersonal version of the problem. (b) For moral dilemmas involving the pandemic, two dimensions were sufficient to capture the variance in the large sample ($N > 15K$) across the 5 dilemmas. The first dimension relates to concern about human life expectancy, whereas the second dimension relates to equity in health, both across people and even across species. (c) These two dimensions were found to be related to the two different underlying dimensions of utilitarian thinking. These findings were replicated in a preregistered study using a representative American sample.

In addition, an examination of individual-differences variables and demographic variables (e.g., severity of the crisis) revealed a number of interesting findings. For example, the severity of the pandemic was associated with endorsement of the “maximizing life expectancy” options whereas there was no correlation with the “equitable public health” dilemmas.

Turning to evaluation, I reviewed the first submission of the paper and was positively impressed at the time but had a few minor comments and one major concern relating to the identifiability of the 2 principal components. The authors have done an outstanding job in their cover letter in addressing this concern, and my detailed reading of the revision has uncovered no further problems. Quite on the contrary, the additional material that has been added to the revision has further improved the quality of the paper, and I am now pleased to recommend publication.

Decision letter (RSOS-210096.R1)

Dear Dr Navajas

On behalf of the Editors, we are pleased to inform you that your Manuscript RSOS-210096.R1 "Moral responses to the COVID-19 crisis" has been accepted for publication in Royal Society Open Science subject to minor revision in accordance with the referees' reports. Please find the referees' comments along with any feedback from the Editors below my signature.

Please submit your revised manuscript and required files (see below) no later than 7 days from today's (ie 25-Aug-2021) date. Note: the ScholarOne system will 'lock' if submission of the revision is attempted 7 or more days after the deadline. If you do not think you will be able to meet this deadline please contact the editorial office immediately.

on behalf of Dr Giorgia Silani (Associate Editor) and Essi Viding (Subject Editor)
openscience@royalsociety.org

Associate Editor Comments to Author (Dr Giorgia Silani):

Associate Editor: 1

Comments to the Author:

The manuscript have been now reviewed by the two previous reviewers. Both recommend acceptance pending minor revisions. You will find their comment attached. I would be happy to proceed without sending it out again, upon clarification of the above mentioned points.

Reviewer comments to Author:

Reviewer: 2

Comments to the Author(s)

Review of MS RSOS-210096.R1

by Navajas et al.

Reviewer: Stephan Lewandowsky

Summary and Overall Recommendation

The paper investigates how people process moral dilemmas relating to the COVID-19 pandemic. The pandemic opened up a slew of moral dilemmas relating to privacy, decisions about rationing medical care, and trading off the economy against public health. The authors ground their work in the classical distinction between utilitarian and deontological principles of morality, and they seek to tease apart those two dimensions by using 5 different dilemmas that are expected to elicit different response patterns depending on which deontological or utilitarian maxim (maximizing life expectancy vs. equitable public health) people choose to satisfy.

The results show (a) that in the conventional trolley problem questionnaire responses that correspond to those two dimensions are independent and can be related to the observed acceptability of the personal and impersonal version of the problem. (b) For moral dilemmas involving the pandemic, two dimensions were sufficient to capture the variance in the large sample (N>15K) across the 5 dilemmas. The first dimension relates to concern about human life expectancy, whereas the second dimension relates to equity in health, both across people and even across species. (c) These two dimensions were found to be related to the two different underlying dimensions of utilitarian thinking. These findings were replicated in a preregistered study using a representative American sample.

In addition, an examination of individual-differences variables and demographic variables (e.g., severity of the crisis) revealed a number of interesting findings. For example, the severity of the pandemic was associated with endorsement of the “maximizing life expectancy” options whereas there was no correlation with the “equitable public health” dilemmas.

Turning to evaluation, I reviewed the first submission of the paper and was positively impressed at the time but had a few minor comments and one major concern relating to the identifiability of the 2 principal components. The authors have done an outstanding job in their cover letter in addressing this concern, and my detailed reading of the revision has uncovered no further problems. Quite on the contrary, the additional material that has been added to the revision has further improved the quality of the paper, and I am now pleased to recommend publication.

Reviewer: 1

Comments to the Author(s)

This paper, *Moral responses to the COVID-19 crisis*, presents a revised version of a manuscript examining how the OUS predicts responses to COVID dilemmas. I already saw much to like about the previous version of the manuscript, and I continue to do so. My concerns were primarily with the conceptual framing of the argument. Here I commend the authors for their thoughtful and thorough response to these concerns, as well as the concerns of the other reviewers. I find this version of the manuscript much improved, with much clearer terminology about decisions versus process, labeling people as decision-makers, and distinguishing between increased utilitarian and reduced deontological decision-making. I think a version of the manuscript very close to this one should be published, pending slight clarification, and I do not think I need to personally see the revision.

The authors write that “we now explicitly state that our work is silent regarding mechanisms that reflect deontological judgements.” While I appreciate the spirit of this argument, I suggest it is not entirely correct. Insofar as the authors examine dilemmas that arguably pit deontological against utilitarian concerns, responses arguably are exactly as informative about deontological as they are of utilitarian responding. In other words, any argument the authors could make that X increases utilitarian responding could simply be rephrased as ‘X reduced deontological responding.’ No meaningful distinction can be made between these statements without use of modelling approaches such as process dissociation or the CNI model. It might be helpful to slightly reword the caveats the authors have helpfully added to clarify this point.

Note this argument primarily applies to the portion of the results pertaining to ‘trolley dilemma’ responses. Arguably the OUS is designed to capture ‘utilitarian thinking’ (although there can be other interpretations as I noted in my previous review). Insofar as this is true, it is reasonable to argue that the authors are examining how ‘utilitarian thinking’ influences COVID responding. So it’s a slightly nuanced point, but arguably the dilemma responses could be reworded as assessing deontological responding and flipping the sign of the correlations and remain equally correct interpretations of the data. Note as well that I say ‘deontological responding’ here as the authors are correct that raw dilemma responses are ambiguous whether they truly assess either utilitarian or deontological ‘thinking’.

Overall, I comment the authors on a fine piece here. Finally, I must apologize as my review likely delayed their response from the editor. Unfortunately, this review found me in the midst of an international move which was especially complicated by COVID restrictions and further complicated by unexpected government problems requiring many hours sorting out immigration status and then active COVID cases in the household... so my reviewing efficiency has fallen off a cliff.

===PREPARING YOUR MANUSCRIPT===

one version identifying all the changes that have been made (for instance, in coloured highlight, in bold text, or tracked changes);
 a 'clean' version of the new manuscript that incorporates the changes made, but does not highlight them. This version will be used for typesetting.

===PREPARING YOUR REVISION IN SCHOLARONE===

- Any electronic supplementary material (ESM).
- If you are requesting a discretionary waiver for the article processing charge, the waiver form must be included at this step.
- If you are providing image files for potential cover images, please upload these at this step, and inform the editorial office you have done so. You must hold the copyright to any image provided.
- A copy of your point-by-point response to referees and Editors. This will expedite the preparation of your proof.

- Ensure that your data access statement meets the requirements at <https://royalsociety.org/journals/authors/author-guidelines/#data>. You should ensure that you cite the dataset in your reference list. If you have deposited data etc in the Dryad repository, please only include the 'For publication' link at this stage. You should remove the 'For review' link.
- If you are requesting an article processing charge waiver, you must select the relevant waiver option (if requesting a discretionary waiver, the form should have been uploaded at Step 3 'File upload' above).
- If you have uploaded ESM files, please ensure you follow the guidance at <https://royalsociety.org/journals/authors/author-guidelines/#supplementary-material> to include a suitable title and informative caption. An example of appropriate titling and captioning may be found at https://figshare.com/articles/Table_S2_from_Is_there_a_trade-off_between_peak_performance_and_performance_breadth_across_temperatures_for_aerobic_scope_in_teleost_fishes_/3843624.

Author's Response to Decision Letter for (RSOS-210096.R1)

See Appendix B.

Decision letter (RSOS-210096.R2)

Dear Dr Navajas,

It is a pleasure to accept your manuscript entitled "Moral responses to the COVID-19 crisis" in its current form for publication in Royal Society Open Science. The comments of the reviewer(s) who reviewed your manuscript are included at the foot of this letter.

COVID-19 rapid publication process:

We are taking steps to expedite the publication of research relevant to the pandemic. If you wish, you can opt to have your paper published as soon as it is ready, rather than waiting for it to be published the scheduled Wednesday.

This means your paper will not be included in the weekly media round-up which the Society sends to journalists ahead of publication. However, it will still appear in the COVID-19 Publishing Collection which journalists will be directed to each week (<https://royalsocietypublishing.org/topic/special-collections/novel-coronavirus-outbreak>).

If you wish to have your paper considered for immediate publication, or to discuss further, please notify openscience_proofs@royalsociety.org and press@royalsociety.org when you respond to this email.

on behalf of Dr Giorgia Silani (Associate Editor) and Essi Viding (Subject Editor)
openscience@royalsociety.org

Associate Editor Comments to Author (Dr Giorgia Silani):
Associate Editor

Comments to the Author:

The authors have satisfactorily addressed all the comments. I am happy to recommend acceptance as it is

Appendix A

Reviewer: 1

This paper, Moral Reasoning about the COVID-19 Crisis, presents two studies examining how scores on the 2D model of utilitarianism or OUS scale predict patterns of responses to covid dilemmas pitting different targets against one another. Overall, there is a lot to like about this paper. The topic is important and timely. The dilemmas are novel and interesting. The authors used a large sample and then a pre-registered replication. The findings are rather clear and make sense considering theory. Overall, I am positively disposed towards this paper and I would like to see a version published. My primary concerns are not with the core work but rather with the conceptual framing the authors have used. My hope is that with modest refinements to the phrasing in the intro and discussion this version could be published.

We thank the referee for thoroughly reviewing our paper and for her/his constructive feedback. We are pleased to read that the reviewer found our work to be timely and important. Based on these comments, we have re-written several key parts of the paper, introducing several refinements to the phrasing of the manuscript. This led to changes in all sections, but most notably in the title, abstract, introduction, and figures 1 and 2.

We believe that the changes suggested by the reviewer have substantially improved the theoretical clarity of our contribution and hence improved the quality of the manuscript. Below, we provide point-by-point responses to each comment.

1) My theoretical concerns are several. First, the authors use the term 'moral reasoning' and claim they are studying 'reasoning mechanisms.' They're not the only ones in the field to use this type of terminology but it is incorrect and should be updated. This terminology comes from the philosophical origins of dilemma studies where philosophers assume that they are studying 'reasoning.' However, this approach assumes the mechanism rather than tests it, and appears to deny the possibility of humans making any decisions that don't involve reasoning. Many theorists describe mechanisms other than reasoning which influence moral judgments (e.g., Haidt, 2001; Greene, 2007) and researchers should not assume that they know what the mechanism is. Instead, I recommend using mechanism-neutral language. What they are really studying here is judgments. Whether such judgments result from reasoning or some other process or a combination of processes is a complex topic that is beyond the scope of this paper. By changing the term to moral judgments or responses or decisions, the authors can neatly sidestep the thorny question of whether it's truly reasoning or some other mechanism involved in driving dilemma responses.

The reviewer is indeed correct that this paper cannot directly speak about the processes underlying moral responses as we cannot rule out several other mechanisms that could contribute to moral decisions. As correctly pointed out by the reviewer, assuming one knows the mechanism based on how people respond to moral scenarios is standard in the literature. However, this is not accurate and certainly here we cannot discard the possibility that emotion rather than reason (or any other process) played an important role in our data.

Based on the reviewer's comment, we removed all words such as "reasoning", "argument", and "mechanism" and instead we now have re-written the paper using mechanism-neutral language (i.e., "judgements", "responses", or "decisions").

This led to several changes spread throughout the manuscript, but the most important ones appear in the title, abstract, Figure 1B, and the introduction.

2) Next, the authors describe what they call a 'unidimensional' model of moral judgment and they reference a paper by Joshua Greene. This is a strange argument to make because Greene's model is famously described as a dual-process model. The authors appear to describe judgments on these dilemmas as reflecting either greater or lesser utilitarian concerns as if there's no mechanism motivating deontological judgments. However, the core of this theory always was that one mechanism (e.g., reasoning about outcomes) motivates utilitarian judgments and a different mechanism (e.g., emotional aversion to harm) motivates deontological judgments. The model specifies that these two mechanisms clash and compete with one another to produce the ultimate response--hence, increasing emotionality can theoretically reduce willingness to cause sacrificial harm without changing the utilitarian outcomes of harm; likewise, increasing utilitarian outcomes of harm can theoretically increase sacrificial willingness even for the same aversive harmful action. The current paper is silent regarding the role of emotion or motivation to avoid causing harm, and this absence is misleading for a field that was founded on a dual-process model which is widely described and discussed, with substantial evidence (albeit many criticisms as well). I am not saying the authors must add additional mechanisms to their studies, but they should clarify the way they describe this model.

The reviewer makes a valid point about our previous reference to a paper by Greene and colleagues. Our aim was explaining that the theory developed in that paper (Dual-process model) proposes the existence of only one kind of mechanism for utilitarian decisions (i.e., reasoning about outcomes) unlike some recent theories (e.g., Kahane et al., 2018) which posit that utilitarian decisions could stem from two different considerations: a high tolerance to instrumental harm and/or a great concern for impartial beneficence.

However, in light of this comment, we agree with the reviewer that referring to dual-process models when we argue that responses may be unidimensional could bring confusion to our readership. So, based on the feedback received from the reviewer, we removed from the introduction all references to dual-process models. Instead, we simply mention that utilitarian decisions could stem from one or two different dimensions and that we empirically tested those competing ideas.

The key paragraph (page 4) now reads:

“We ground our work on the study of utilitarian decision-making (21). Utilitarianism is a normative theory that advocates that decisions ought to maximize a utility function reflecting the happiness and well-being of all affected individuals. As an example, in trolley-like dilemmas where different decisions lead to outcomes that vary in the number of deaths, the utilitarian prescription is to select the option in which less people die. More recently, it has been argued that people making utilitarian decisions in trolley-type dilemmas may in fact be relying on different considerations that produce the same outcome (22). According to this view, utilitarian preferences result from two very distinct aspects: a ‘negative’ dimension that reflects a permissive attitude towards instrumental harm, and a ‘positive’ dimension, called impartial beneficence, which reflects an unbiased concern for the wellbeing of all sentient lives. Previous research has shown that, while instrumental harm correlates with psychopathic tendencies, impartial beneficence is associated with higher empathic concern (22).”

3) Part of the confusion may stem from the authors use of the OUS/2D utilitarian paper. Notably, that paper only describes mechanisms pertaining to utilitarian decision making and is silent regarding mechanisms that reflect deontological decision making--i.e., this approach is only focused on one of the two processes described by the dual-process model, and is silent regarding motivations to reject causing harm (i.e., actively make deontological judgments rather than simply avoid utilitarian judgments). In other words, the original model in the field is a dual-process model, and the 2D model further suggests that one of those processes may be further subdivided into more than one type of utilitarian consideration, but those does not mean that the other theorized process does not exist or influence judgments in its own right. Describing traditional models as unidimensional appears to miss a large and growing literature that uses processing models to assess responses. The most prominent examples of this literature include process dissociation (Conway & Gawronski, 2013) and the CNI model (Gawronski et al, 2017). Papers using these techniques clearly demonstrate that more than one dimension contributes to what the authors here describe as a uni-dimensional model. In other words, what the authors describe as a single dimension with utilitarianism at one extreme and deontologism at the other extreme is better described as the combination of one response tendencies to reject causing harm or follow moral rules and another response tendency to maximize overall outcomes-- both elements in the ous pertain only to the second dimension reflecting concern for outcomes. No dimension of the ous maps theoretically onto deontological concerns about rejecting harm or consistently following moral rules.

I suggest the authors clarify these theory points as the way the paper is currently written may be confusing to readers. This paper is silent on the possibility that agreement with or endorsement of deontological considerations independent of the absence of utilitarian considerations may also play a role in driving judgments, and this point should be clarified for people who are familiar with the dual-process model and may wonder about the absence of reference to the second process in this model. Certainly, the authors should modify language about a uni-dimensional model as that does not accurately describe the model they refer to.

Based on the feedback received form the reviewer, we removed all parts of the paper suggesting that dual-process models are “unidimensional”. Furthermore, we now explicitly state that our work is silent regarding mechanisms that reflect deontological judgements. We also agree that our focus on instrumental harm and impartial beneficence does not mean that other theorized process does not exist or influence judgments above and beyond the effects described in this paper. These modifications now appear in a new paragraph in the Discussion (page 22) that reads:

“While this paper focuses on the relationship between responses to moral problems of the COVID-19 crisis and utilitarian decision-making, we acknowledge that our work is silent about other possible mechanisms such as emotion or motivation to avoid harm (38,39). This focus on utilitarian decisions does not mean that other theorized process does not exist or influence judgments above and beyond the observed effects (40,41). Moreover, this research does not suggest that scoring lowly on instrumental harm or impartial beneficence reflects evidence for engaging in other kind of reasoning processes such as deontological thinking (42). In other words, the fact that we found a correlation between utilitarian judgements and prioritizing public health does not imply that people making deontological decisions are less concerned about public health during the pandemic. Future research should explore how the endorsement of deontological considerations (43) relate to moral responses about this healthcare crisis.”

4) Another point that could stand clarification is the way the authors describe utilitarian and deontological principles in connection to the dilemmas in the paper. They do a nice job of briefly describing the relevant philosophical positions, but it should be clarified that responses on dilemmas should be conceptualized as distinct from endorsement of philosophical principles. In dilemmas, participants are making decisions and theorists may describe as consistent with utilitarian or philosophical principles, but it is not correct to say that the people making the decisions are necessarily invoking or endorsing those principles in the making of the decision. Hence, making a decision to accept sacrificial harm does not guarantee that one should be described as a utilitarian, as there are many reasons one may arrive at that judgment. Nor does making a judgment on one dilemma guarantee that one will make a similar judgment across many dilemmas, although there are somewhat stable individual differences. Likewise, one cannot determine whether someone counts as a deontologist or not based on responses to the two-dimensional OUS scale or dilemma responses, as there can be many factors that contribute to self-report responses--for example, psychopathy also predicts higher responses on instrumental harm, so a person endorsing this dimension may not necessarily qualify as 'utilitarian' even though some utilitarian philosophers also endorse the same items.

Conway and colleagues 2018 argued that the terms utilitarian and deontology better describe the decisions themselves rather than the people making those decisions, and recommend researchers avoid describing people as 'utilitarians or deontologists' except in rare cases involving practicing philosophers who fully appreciate and understand what it means to endorse these philosophical notions--it is more accurate to say 'people who make utilitarian judgments' or 'people accepting sacrificial harm' or 'utilitarian decision-makers' which implies the decision may be utilitarian without affirming the person as necessarily so. From this perspective, Figure 2 is misleading because it uses the terms 'utilitarians' and 'deontologists.'

We agree with the point made by the reviewer in this comment. We were unaware of the sound argument made by Conway and colleagues (2018) and that it was misleading to classify participants as “utilitarians” or “deontologists”. In response to this observation, we have modified the wording in the entire paper and no longer describe individuals in that way. Instead, we now refer to “people making utilitarian decisions” or “utilitarian decision-makers”. We also cite the work by Conway and colleagues when we explicitly avoid classifying people based on the responses obtained in this work:

“Therefore, seeing that instrumental harm or impartial beneficence correlates with the two principal components observed in this study does not necessarily imply that participants always rely on those arguments across all scenarios or that the OUS scale will explain moral decisions in a wider set of dilemmas about the COVID-19 crisis (42).”

Importantly, we have now removed the panel of Figure 2 where we classified individuals based on the OUS score. While the terminology of that classification was not original from this study but proposed in Kahane et al. (2018), we agree with the reviewer that it is inaccurate to assume that people scoring lowly on the OUS endorse deontological principles. This is now explicitly stated in page 22:

“Moreover, this research does not suggest that scoring lowly on instrumental harm or impartial beneficence reflects evidence for engaging in other kind of reasoning processes such as deontological thinking (42).”

5) In the case of the OUS, it may be reasonable to describe people as endorsing utilitarian principles, but that still doesn't mean that people should be described as 'utilitarians' full stop. People may endorse principles for a variety of reasons and the reasons they have as laypeople may not necessarily map onto the reasons why utilitarian expert philosophers endorse the same principles. The tendency to describe people as utilitarian or deontological obscures the true complexity of the many psychological processes contributing to judgments in this area. People arrive at judgments through a multiplicity of processes (including two studies in the current work, but of course the current work omits measuring many other processes demonstrated in other work, such as affective reactions to harm, concerns about moral rules, general inaction, self-presentation, etc.).

Moreover, I don't think it's correct to describe people as 'deontologists' if they score low on one or both dimensions of the OUS. Instead, one would need a different measure asking questions directly about deontological ethics and how much people endorse those (e.g., see Robinson et al., 2015 who examine self-report items separately assessing utilitarian and deontological considerations). Failing to endorse utilitarian principles is not the same thing as definitely endorsing deontological principles.

Here, we also agree with the reviewer that the previous manuscript fell short in explaining that our work studies the relationship between moral responses to the pandemic and utilitarian decision-making while it is silent about other processes that could inform moral judgements such as deontological considerations. To clearly communicate that people may arrive at moral judgements through different mechanisms, we now explicitly state so in the Discussion (pages 22 and 23). The key paragraph now reads:

“This research aimed at studying inter-individual differences in moral decision-making during the pandemic. We show that people who support utilitarian decisions in the trolley dilemma (1) and in the scenarios constituting the OUS (22) concurrently make a series of acceptability judgements in moral scenarios about the COVID-19 crisis. However, lay people arrive at judgments through a multiplicity of processes which may or may not coincide with the utilitarian principles proposed by theorists. Therefore, seeing that instrumental harm or impartial beneficence correlates with the two principal components observed in this study does not necessarily imply that participants always rely on those arguments across all scenarios or that the OUS scale will explain moral decisions in a wider set of dilemmas about the COVID-19 crisis (42).”

We also removed from the main text all parts which seemed to suggest that participants endorsed deontological principles because they did not make utilitarian judgements. As an example of how future work could study their relationship with moral judgements during the pandemic, we cite the work by Robinson et al. (2015):

“Future research should explore how the endorsement of deontological considerations relate to moral responses about this healthcare crisis.” (page 22)

Finally, and following the reviewer's comment, we now explicitly state that scoring lowly on the OUS does not necessarily imply that people engage in deontological thinking (page 22):

“Moreover, this research does not suggest that scoring lowly on instrumental harm or impartial beneficence reflects evidence for engaging in other kind of reasoning processes such as deontological thinking (42).”

6) Although an important theory issue for clarity, in a way these are also fairly modest wording issues and I hope the authors can simply tweak their language slightly without changing much of the core of the paper. The actual core contribution of this paper which is to distinguish between two different aspects of utilitarian thinking, to either maximize overall life expectancy or equitability in health treatment. This is where the main focus should lie, and cleaning up the theory language on the way to making this point will help the reader focus on and appreciate this main contribution to the literature rather than getting hung up on confusion regarding the way past work is conceptualized. I think it is interesting that impartial beneficence and instrumental harm demonstrate different relationships with maximizing life expectancy and equitable public healthcare access, and recommend the authors try to focus the discussion on this point itself without trying to make too many broader points about the overall nature of human moral decision-making given that doing so would require engaging with a much vaster literature than the current paper does--i.e., these findings do not rule out the presence or importance of many other factors such as concern for rules or affective decision-making and so on, they simply demonstrate the presence of two somewhat distinct influences of the 2D scale. I buy these findings, they make sense and the analyses the author's present are convincing.

We thank the reviewer for the constructive feedback provided in this letter. In the revised version of the manuscript, we focused as much as possible on the core contribution of the paper which distinguishes different aspects of utilitarian decision-making and their relationship with moral responses to the pandemic. We substantially changed several aspects from the introduction and discussion being careful to not make many broader claims about the overall nature of moral decision-making. We also explicitly state that our work does not rule out the importance of factors that remained unexplored in this study, such as the role of emotion or deontological considerations.

Based on these comments and the refinement of the language employed in the paper, we believe that our work has been substantially improved and so we wholeheartedly thank the reviewer for her/his feedback.

Reviewer: 2

The paper investigates how people process moral dilemmas relating to the COVID-19 pandemic. The pandemic opened up a slew of moral dilemmas relating to privacy, decisions about rationing medical care, and trading off the economy against public health. The authors ground their work in the classical distinction between utilitarian and deontological principles of morality, and they seek to tease apart those two dimensions by using 5 different dilemmas that are expected to elicit different response patterns depending on which deontological or utilitarian maxim (maximizing life expectancy vs. equitable public health) people choose to satisfy.

The results show (a) that in the conventional trolley problem questionnaire responses that correspond to those two dimensions are independent and can be related to the observed acceptability of the personal and impersonal version of the problem. (b) For moral dilemmas involving the pandemic, two dimensions were sufficient to capture the variance in the large sample ($N > 15K$) across the 5 dilemmas. The first dimension relates to concern about human life expectancy, whereas the second dimension relates to equity in health, both across people and even across species. (c) These two dimensions were found to be related to the two different underlying dimensions of utilitarian thinking. These findings were replicated in a preregistered study using a representative American sample.

In addition, an examination of individual-differences variables and demographic variables (e.g., severity of the crisis) revealed a number of interesting findings. For example, the severity of the pandemic was associated with endorsement of the “maximizing life expectancy” options whereas there was no correlation with the “equitable public health” dilemmas.

Turning to evaluation, there is much to like about the paper. It is generally well written, concise, easy to follow, and highly informative. The method is innovative but also very solid, and the data are analyzed competently and in great depth.

On balance, this paper makes a very interesting contribution and I am positively inclined towards publication. However, I believe that there is one major question that needs to be resolved by revision before I can wholeheartedly recommend publication. (I should add that I rarely only pick up on a single issue, so the authors are to be commended for submitting a terrific paper that is already in very good shape).

We thank the referee for thoughtfully reviewing our work. We are pleased to read that the reviewer liked the paper and thank him for the positive assessment of the manuscript. Below, we provide a detailed response to the major question raised in this review. We believe that the robustness analysis suggested by the referee, which is now included in the main text (page 12) and as a supplementary figure (Fig. S4), brings greater assurance to our readership and hence increased the quality of the paper.

Major point

My biggest concern involves the identifiability of the two principal components, given that the first 3 dilemmas load positively on both. How robust are the observed findings given that ultimately only two dilemmas carry much of the diagnostic load? How robust are the findings in light of this? I am prepared to accept the results as presented, but I would ask for some assurance that there are no surprises lurking in the shadows. I'll leave it up to the authors to decide how to respond to this but I suspect most readers would like some more assurance.

The reviewer raised a valid concern about the identifiability of the two principal components. This paper analysed acceptability judgements across five dilemmas, defining a native 5-dimensional space, and we found that only two principal components explained a significant proportion of the variance in the data. As the reviewer correctly points out, two dilemmas have different signs in their loadings to each component: the tension between saving younger people and treating all patients equally (dilemma #4) and between accelerating vaccine development and respecting animal rights (dilemma #5).

Consequently, one could reasonably ask whether our findings are robust given that only two dilemmas carry much of the diagnostic load. To answer this question, we performed a robustness check by which we sequentially removed data from single dilemmas. By this, we sought to establish whether our main findings were driven by the observations made in the dilemma that was excluded from the analysis.

To this end, we performed a sequential procedure that replaced data from single dilemmas with random uniform noise while not modifying the observations from the remaining four scenarios. We repeated this procedure 10,000 times for each dilemma and ran the same analyses reported in the paper using these synthetic datasets.

We first asked whether two principal components still explained a significant proportion of the variance in the simulated data. The figure below shows the fraction of the variance explained by the second principal component when we removed each dilemma from our dataset and replaced it with random noise. We observed that the second principal component still explained a greater-than-chance proportion of the variance suggesting that the bi-dimensional organization of acceptability responses was not driven by any single dilemma. As a control (grey dot), we verified that when we simultaneously replaced all observations with random noise, then we see that the second component explains a fraction of the variance which is consistent with chance.

This analysis also revealed that the tension between accelerating vaccine development and respecting animal rights (dilemma #5) was more important in eliciting two-dimensional responses than the one between saving younger people and treating all patients equally (dilemma #4).

Then, we addressed the key question made by the reviewer which is whether we could still identify those two principal components as “human life expectancy” (positive loadings for all dilemmas) and “equitable public health” (positive loadings for the first three dilemmas and negative loadings for the last two).

The figure below shows the median loadings obtained in our simulations when we removed each dilemma. This shows that the sign of the loadings in the first two principal components were, in all cases, consistent with a concern about “human life expectancy” (first column) and “equitable public health” (second column).

Overall, we believe that this analysis provides greater assurance to our readership about the robustness of our findings. These observations suggest that the identifiability of two principal components that reflect “human life expectancy” and “equitable public health” are not entirely driven by the observations made in any single dilemma but reflect a strong interplay of correlations across scenarios. The results obtained in Study 2 indicate that this result is not only robust but also replicable in a different setting.

We thank the reviewer for suggesting this important robustness check, which is now reported in page 12 and Figure S4.

Appendix B

Associate Editor Comments to Author (Dr Giorgia Silani):

Associate Editor: 1

Comments to the Author:

The manuscript have been now reviewed by the two previous reviewers. Both recommend acceptance pending minor revisions. You will find their comment attached. I would be happy to proceed without sending it out again, upon clarification of the above mentioned points.

We thank the editor for her thoughtful feedback and for accepting our manuscript subject to minor modifications. We hereby summarize the changes made in this submission:

- **Reviewer #2 suggested no changes and recommended publication in its current form.**
- **As requested by Reviewer #1, we have now reworded one paragraph in the Discussion (page 22)**

The key paragraph is highlighted in red in the main manuscript. It now reads:

While this paper focuses on the relationship between responses to moral problems of the COVID-19 crisis and utilitarian judgements, this does not mean that other theorized processes (for example, deontological mechanisms based on emotion or motivation to avoid harm (38,39)) does not exist or influence judgments above and beyond the observed effects (40,41). Future research should explore how the endorsement of deontological considerations relate to moral responses about this healthcare crisis (42, 43).

Below, we provide point-by-point responses to both reviewers' comments and we wholeheartedly thank the editor for potentially proceeding with acceptance of this manuscript without sending it out for peer-review.

Reviewer: 2

Reviewer: Stephan Lewandowsky

Summary and Overall Recommendation

The paper investigates how people process moral dilemmas relating to the COVID-19 pandemic. The pandemic opened up a slew of moral dilemmas relating to privacy, decisions about rationing medical care, and trading off the economy against public health. The authors ground their work in the classical distinction between utilitarian and deontological principles of morality, and they seek to tease apart those two dimensions by using 5 different dilemmas that are expected to elicit different response patterns depending on which deontological or utilitarian maxim (maximizing life expectancy vs. equitable public health) people choose to satisfy.

The results show (a) that in the conventional trolley problem questionnaire responses that correspond to those two dimensions are independent and can be related to the observed acceptability of the personal and impersonal version of the problem. (b) For moral dilemmas involving the pandemic, two dimensions were sufficient to capture the variance in the large sample ($N > 15K$) across the 5 dilemmas. The first dimension relates to concern about human life expectancy, whereas the second dimension relates to equity in health, both across people and even across species. (c) These two dimensions were found to be related to the two different underlying dimensions of utilitarian thinking. These findings were replicated in a preregistered study using a representative American sample.

In addition, an examination of individual-differences variables and demographic variables (e.g., severity of the crisis) revealed a number of interesting findings. For example, the severity of the pandemic was associated with endorsement of the “maximizing life expectancy” options whereas there was no correlation with the “equitable public health” dilemmas.

Turning to evaluation, I reviewed the first submission of the paper and was positively impressed at the time but had a few minor comments and one major concern relating to the identifiability of the 2 principal components. The authors have done an outstanding job in their cover letter in addressing this concern, and my detailed reading of the revision has uncovered no further problems. Quite on the contrary, the additional material that has been added to the revision has further improved the quality of the paper, and I am now pleased to recommend publication.

We thank the referee for reviewing this version of the manuscript and for his positive assessment of our work. We are glad that the reviewer found no further problems in our work and that he recommended publication of this paper.

Reviewer: 1

Comments to the Author(s)

This paper, Moral responses to the COVID-19 crisis, presents a revised version of a manuscript examining how the OUS predicts responses to COVID dilemmas. I already saw much to like about the previous version of the manuscript, and I continue to do so. My concerns were primarily with the conceptual framing of the argument. Here I commend the authors for their thoughtful and thorough response to these concerns, as well as the concerns of the other reviewers. I find this version of the manuscript much improved, with much clearer terminology about decisions versus process, labeling people as decision-makers, and distinguishing between increased utilitarian and reduced deontological decision-making. I think a version of the manuscript very close to this one should be published, pending slight clarification, and I do not think I need to personally see the revision.

We thank the reviewer for the constructive feedback which helped us substantially improving this manuscript. Below, we provide a response to the comment raised by the referee.

The authors write that “we now explicitly state that our work is silent regarding mechanisms that reflect deontological judgements.” While I appreciate the spirit of this argument, I suggest it is not entirely correct. Insofar as the authors examine dilemmas that arguably pit deontological against utilitarian concerns, responses arguably are exactly as informative about deontological as they are of utilitarian responding. In other words, any argument the authors could make that X increases utilitarian responding could simply be rephrased as ‘X reduced deontological responding.’ No meaningful distinction can be made between these statements without use of modelling approaches such as process dissociation or the CNI model. It might be helpful to slightly reword the caveats the authors have helpfully added to clarify this point.

Note this argument primarily applies to the portion of the results pertaining to ‘trolley dilemma’ responses. Arguably the OUS is designed to capture ‘utilitarian thinking’ (although there can be other interpretations as I noted in my previous review). Insofar as this is true, it is reasonable to argue that the authors are examining how ‘utilitarian thinking’ influences COVID responding. So it’s a slightly nuanced point, but arguably the dilemma responses could be reworded as assessing deontological responding and flipping the sign of the correlations and remain equally correct interpretations of the data. Note as well that I say ‘deontological responding’ here as the authors are correct that raw dilemma responses are ambiguous whether they truly assess either utilitarian or deontological ‘thinking’.

We agree with the referee that no meaningful distinction can be made between increased utilitarian decision-making and decreased deontological decision-making. Based on this feedback, we have now reworded the paragraph in the Discussion where we referred to these thoughts. In particular, we have removed the sentence where we said that “our work is silent about mechanisms that reflect deontological judgements”. The key paragraph now reads:

While this paper focuses on the relationship between responses to moral problems of the COVID-19 crisis and utilitarian judgements, this does not mean that other theorized processes (for example, deontological mechanisms based on emotion or motivation to avoid harm (38,39)) does not exist or influence judgments above and

beyond the observed effects (40,41). Future research should explore how the endorsement of deontological considerations relate to moral responses about this healthcare crisis (42, 43).

Overall, I comment the authors on a fine piece here. Finally, I must apologize as my review likely delayed their response from the editor. Unfortunately, this review found me in the midst of an international move which was especially complicated by COVID restrictions and further complicated by unexpected government problems requiring many hours sorting out immigration status and then active COVID cases in the household... so my reviewing efficiency has fallen off a cliff.

We thank again the reviewer for providing us with valuable feedback in these difficult times.